

# Modelling the effectiveness of GLOF DRM measures - a case study from the Ala-Archa valley, Kyrgyz Republic

Laura Niggli[1], Holger Frey[1], Simon Allen[1,2], Nazgul Alybaeva[3], Christian Huggel[1], Bolot Moldobekov[3], Vitalii Zaginaev[4]

[1]University of Zurich, Department of Geography, Switzerland
[2]University of Geneva, Switzerland
[3]Central-Asian Institute for Applied Geosciences (CAIAG), Kyrgyz Republic
[4]University of Central Asia, Mountain Societies Research Institute, Kyrgyz Republic

*Correspondence to*: Laura Niggli ([laura.niggli@geo.uzh.ch](mailto:laura.niggli@geo.uzh.ch))

**Abstract.** Disaster risk management (DRM) for glacial lake outburst floods (GLOFs) is critical due to the increasing risk posed by GLOFs to downstream communities and infrastructure. However, the effectiveness of DRM measures remains insufficiently understood, which hinders effective and target-oriented decision-making in GLOF DRM. Existing research predominantly focuses on hazard aspects, with few scientific studies modelling the impacts of DRM measures comprehensively. In order to fill this gap, this study assesses the effectiveness of three different DRM measures for GLOFs in the Ala-Archa catchment, Kyrgyz Republic. Using numerical modelling, we map and assess the effect of three DRM measures: lake lowering, a deflection dam, and a retention basin and compare it to the current baseline hazard map. We develop a hazard reduction score for comparison of the measures and evaluate their effectiveness based on cost and benefit considerations. This study proposes a conceptual framework and methodology that can guide the management of GLOF and debris flow risks in similar contexts globally.

## 1 Introduction

Glacial lakes can represent a substantial hazard in the form of glacial lake outburst floods (GLOFs). A GLOF is the catastrophic release of large amounts of water from a lake that has formed either in front, at the side, within, beneath or on the surface of a glacier (GAPHAZ, 2017). Such mass movements are difficult to predict, can arrive with little warning, cause cascading chains of events and impacts (Mani et al., 2023), and can be very far reaching, highly destructive, and cause extensive loss and damage to property, infrastructure, life and livelihoods (Allen et al., 2016; T. Zhang et al., 2024). Associated with global warming and glacier retreat, the number and size of glacial lakes has rapidly grown since 1990 (Shugar et al., 2020), also leading to changes in GLOF threats and activity (Emmer, 2024; Taylor et al., 2023; G. Zhang et al., 2024). At the same time, infrastructure (e.g., roads, settlements, hydropower plants, etc.) has expanded and human exposure has increased in many mountain regions (Allen et al., 2019; Haeberli et al., 2017; Immerzeel et al., 2020; Schwanghart et al., 2016). GLOF disaster risk management (DRM) has therefore increasingly gained attention. With risk being constituted by the three drivers of hazard (defined as a combination of event likelihood and magnitude), exposure (of any kind of assets including people), and vulnerability (of the affected asset or system) (IPCC, 2018), GLOF risk reduction can be achieved through a reduction in any one or combination of these drivers. Respective DRM measures can have structural and non-structural components, and they can be aimed at a short-, medium- and long-term temporal frames (Niggli et al., 2024). While on a global scale, many different GLOF DRM measures have been implemented and documented (for a global review see Niggli et al. (2024)), few scientific works have modelled the effectiveness of DRM measures (e.g., (Sattar et al., 2023)), and conceptual comparisons of different measures in terms of their benefits and cost are largely missing. While there are DRM evaluation studies in the broader hazard context, for example for debris-flows (Ballesteros Cánovas et al., 2016; Bernard et al., 2019; Chen et al., 2010; Chiou et al., 2015), especially for GLOF risk they mainly focus on the resulting change in hazard (e.g., (Kolenko et al., 2004; Sattar et al., 2023)). DRM generally involves a range of alternative measures rather than a single solution. Cost-benefit analysis (CBA) is a widely used decision-support tool for evaluating and prioritizing these measures by comparing them under a common economic efficiency criterion (e.g., (Benson & Twigg, 2004; Kopp et al., 1997;



Mechler, 2016)). While DRM CBA is a relatively common approach for natural hazards like floods (e.g., (Hudson & Wouter Botzen, 2019; Kull et al., 2013; Rai et al., 2020; Shreve & Kelman, 2014;
Willenbockel, 2011)) and earthquakes (e.g., (Cardona et al., 2008; Dan, 2018; Hoyos & Silva, 2022; Kenny, 2009; Riedel & Guéguen, 2018)), it has not been applied in the context of GLOFs. In theory, for a complete CBA all costs and benefits need to be monetized and aggregated. In DRM, the primary benefits are often the avoided or reduced potential damages and losses (Mechler et al., 2008) and the primary costs are the implementation costs. While many of these damages and losses can be valued in monetary terms,
for example through (avoided) costs of infrastructure reconstruction, there are additional benefits and costs that are less easily quantifiable. The present study models both DRM-induced changes in hazard as well as in exposure. In order to develop a conceptual approach for comparison and evaluation of GLOF DRM policy and decision-making, this study compares different GLOF DRM measures for a GLOF prone catchment in the Central Asian Kyrgyz Range in terms of their quantifiable benefits and cost. It
systematically analyses the pixel-level hazard class change based on hazard maps, as well as the exposure change based on building and tourist area exposure maps for different measures. The hazard changes are weighted and transformed into a hazard reduction score and the exposure change is quantified in terms of cost and benefit allowing for a comparison of the different measures. This case study on the Ala-Archa valley shall serve the purpose of providing a concept for (cost) effectiveness of DRM measures that can
be applied in similar situations to manage GLOF and debris flow risk.

## 2 Study site

The Ala-Archa valley is a partially glaciated catchment of an area of 194 km² and extending over an elevation range of 1500-4895 m.a.s.l., located in the central part of the northern slope of the Kyrgyz Ala-Too, northwestern Tien Shan (Fig. 1). The valley comprises several villages and a national park situated
40 km south of Bishkek, the capital of the Kyrgyz Republic. The National Park is one of the country's most visited places by local and international tourists. There are several lakes and lake complexes in the upper reaches of the Ala-Archa catchment that have acted as sources for GLOFs (Zaginaev, Petrakov, et al., 2019; Zaginaev et al., 2024).

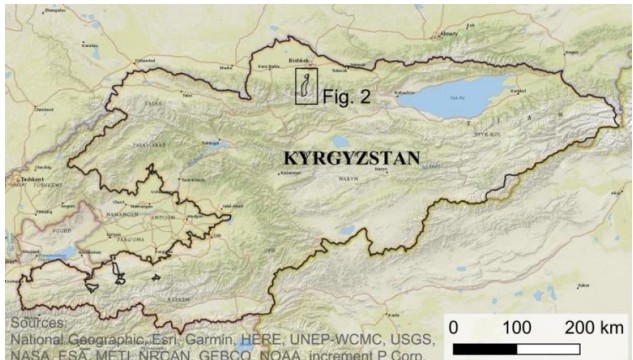

**Figure 1**: Location of the Ala-Archa catchment in Kyrgyzstan. A detailed overview of the catchment is shown in Figure 2.


Two of those lake complexes (Fig. 2) - Teztor and Aksay - have caused repeated GLOFs in the (recent) past and pose a threat up until today. Lake Aksay is an englacial lake (42° 31' 33" N, 74° 31' 56" E) that formed and drained repeatedly in the 1960's (Erokhin & Zaginaev, 2020; Zaginaev et al., 2016) causing
damage downstream and largely being responsible for the formation of the Aksay debris fan, on which most of the National Park tourist infrastructure and leisure area are situated. The currently present neighbouring non-stationary lake Uchitel (42° 31' 40" N, 74° 32' 26" E) is a proglacial lake located in front of the Aksay glacier. It burst in 2015 and caused damage downstream, partially destroying a road and several buildings on the Aksay fan, according to records of the national Ministry of Emergency
Situations (MES).
Teztor lake complex (42° 32' 05" N, 74° 26' 20" E) hosts a number of non-stationary lakes located in a large moraine-glacial complex in the upper reaches of the Adygene valley (left lateral tributary of the Ala-




Archa River). The main Teztor lake has repeatedly caused debris flows (Erokhin & Dikikh, 2003) with the most recent ones in 2005, 2012 and 2018 (Erokhin et al., 2018), of which especially the 2012 event caused damage to buildings and infrastructure in the Ala-Archa valley (Erokhin & Zaginaev, 2020).

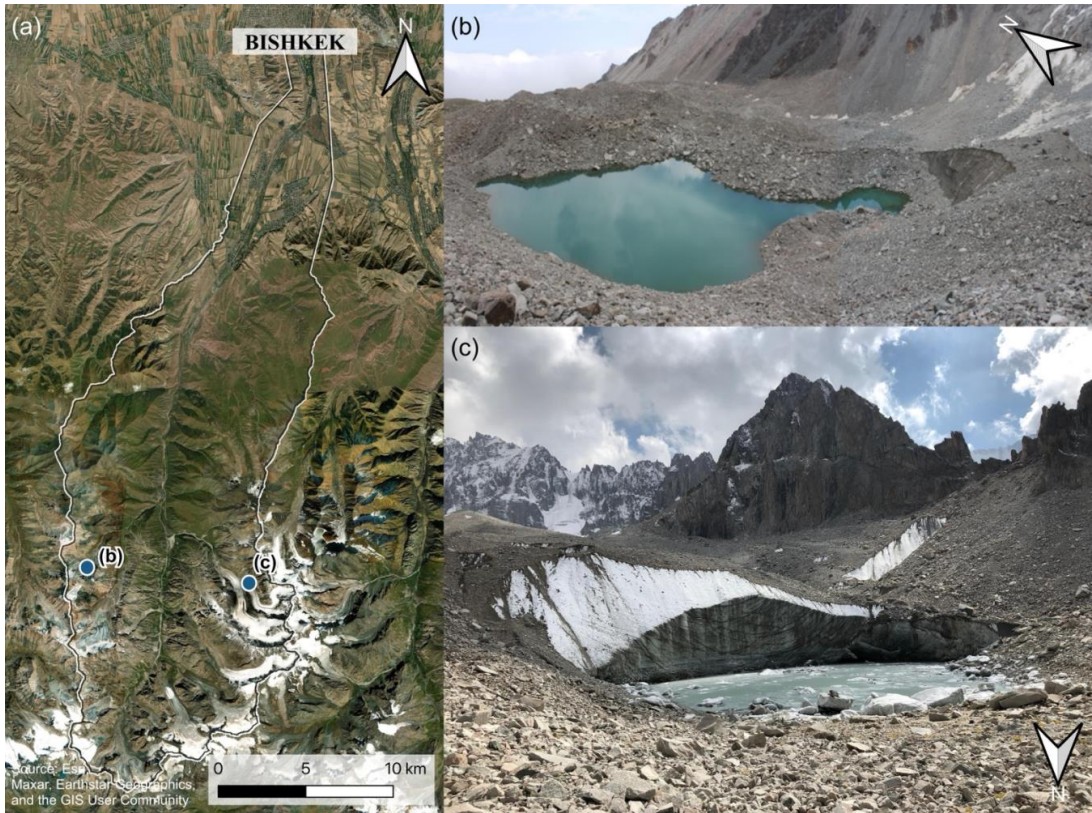

**Figure 2**: **(a)** Satellite image of the Ala-Archa catchment with the location of lakes Teztor (b) and Uchitel (c). **(b)** Photograph of Lake Teztor from July 21st, 2012, 12 days before the outburst (V. Zaginaev). **(c)** Photograph of Lake Uchitel from September 2022 (L. Niggli).

## 3 Methods

### 3.1 Study design

This study compares three different potential GLOF DRM measures in the Ala-Archa catchment (Fig. 3) in terms of effectiveness. The Ala-Archa valley can be seen as representative for many glaciated mountain catchments with presence of lakes, settlements and touristic infrastructure. The effectiveness of three structural measures is assessed in terms of the change of GLOF risk they entail, compared to the hazard map and exposure analysis of a baseline case with no DRM measure in place. GLOF hazard is assessed through mass flow simulations undertaken with the Rapid Mass Movement Simulation (RAMMS) software (Christen et al., 2010) for different GLOF scenarios under the presence and absence of different GLOF DRM measures. The cost-benefit relationship for the three measures is assessed in monetary terms (i.e., cost of averted damage vs. cost of implementation and maintenance of a measure). Four cases are assessed in detail, namely, the baseline case as well as i) partial drainage of the two glacial lakes, ii) a deflection dam close to the tourist area in the upper reaches of the catchment, and iii) a retention basin in the lower catchment.





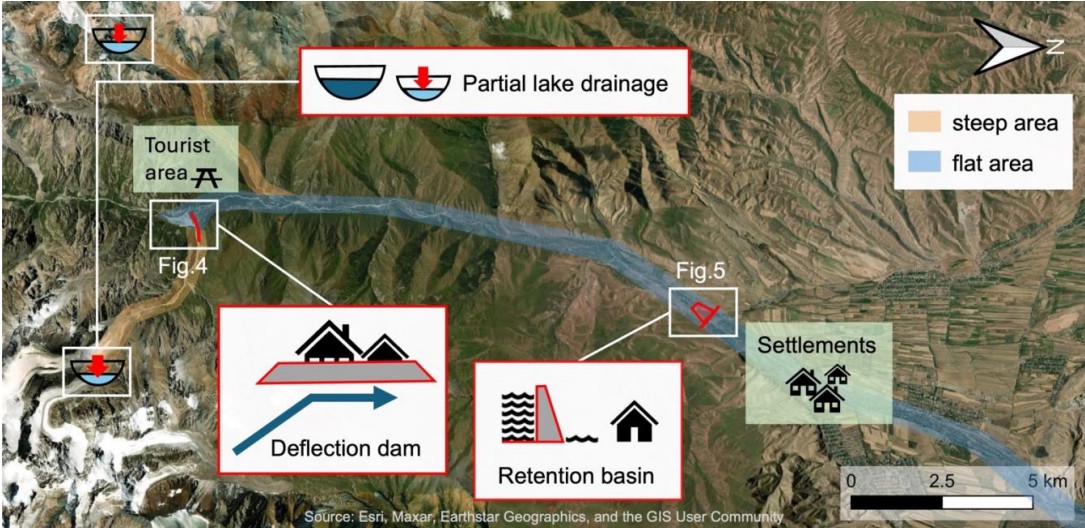

**Figure 3:** Schematic overview of the Ala-Archa valley with the location of the different DRM measures consisting of i) lake level lowering, ii) a deflection dam (reference to Fig. 4), and iii) a retention basin (reference to Fig. 5).

i) GLOF-prone lakes have in the past in various mountain regions been lowered through repeated syphoning and pumping as well as lowered and drained through channel or tunnel excavation (Niggli et al., 2024). Siphoning and pumping usually allow for lake lowering of several meters (Niggli et al., 2024). In this study we simulate a partial lake drainage, lowering the lake levels through syphoning or pumping annually or when needed and reducing the initial water volumes available for GLOFs by 50 %.

ii) For the simulation of a deflection dam, we increased and extended an existing dam above the Aksay fan (Fig. 4). While the existing structure, built between 2013 and 2015, is roughly 400m long and 2m high, we simulate a deflection dam of larger proportions (i.e., 500 m length and 8 m height) at the same location for assessment. These proportions are meant to be understood in a conceptual way rather than from a civil engineering perspective, i.e., structural details of the construction of such a dam are not considered here.

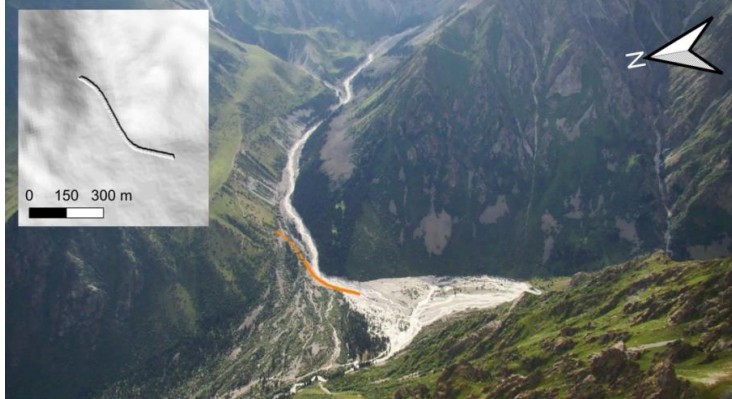

**Figure 4:** Photograph of the Aksay fan with the position of the existing deflection dam as of 2024 (solid orange line) and the enhanced deflection dam (dashed orange line). Box top left: Hillshade of the artificially modified DEM produced for the RAMMS simulation of the DRM measure. Photograph: V. Zaginaev, 2015.

iii) The retention basin and dam DRM measure is based on a present structure located in the village of Kashka Suu (42° 40' 52" N, 74° 31' 01" E) (Fig. 5). The dam (~2-3 m freeboard) and basin were built in the mid 20th century as sediment settling tank and water intake structure for irrigation purposes. The basin was sediment-filled to a large degree in 2024. Results of an aerial survey indicated a usable volume of $2.8 \times 10^4$ m³. For the assessment of such a measure we simulated a deeper basin (~10 m excavation) and a higher dam (~10 m), resulting in a larger retention capacity (area: $1.37 \times 10^5$ m², volume: $1.37 \times 10^6$ m³).




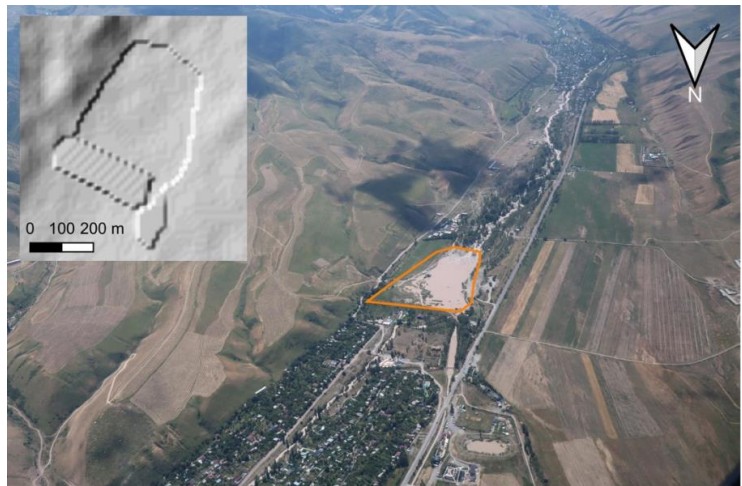

**Figure 5:** Photograph of the dam and retention basin (outlined in orange) in the village of Kashka Suu. Box top left: Hillshade of the artificially modified DEM for the RAMMS simulation of the DRM measure of the retention basin. Photograph: V. Zaginaev, July 2024.

### 3.2 Hazard assessment

In order to assess the baseline GLOF hazard in the Ala-Archa valley (current state, without GLOF DRM measures), we simulated a medium and a large GLOF scenario for the catchment's two most important lake complexes with the debris flow module of the RAMMS software, where the medium and large GLOF scenarios for Uchitel and Teztor lakes are qualitatively assigned medium and low probability levels (after GAPHAZ (2017)). RAMMS is a two-dimensional numerical simulation model that has been extensively

used for simulating avalanches, debris flows and rockfalls (Bartelt et al., 2022; Christen et al., 2010), but also for GLOFs (Frey et al., 2018; Schneider et al., 2014). Table 1 summarizes the most important input parameters and data for the RAMMS simulations of the GLOF scenarios in the baseline case of no implemented DRM measure. A complete list of input data used in all the simulation runs for all DRM measures can be found in the supplementary material.


**Table 1:** Key input parameters and values for the baseline scenarios of GLOFs originating from lake Uchitel or lake Teztor.

|  | **Uchitel lake** | | **Teztor lake** | |
|---|---|---|---|---|
| **Scenario** | medium | large | medium | large |
| **Lake volume** [m³] | 200'000 | 300'000 | 100'000 | 400'000 |
| **Maximum discharge** [m³ s⁻¹] | 350 | 500 | 200 | 650 |
| **Flow density** [kg m⁻³] | 1800/1100 steep/flat | 1800/1100 steep/flat | 1400/1100 steep/flat | 1400/1100 steep/flat |
| **Turbulent friction parameter** ξ [m s⁻²] | 400 | 400 | 400 | 400 |
| **Dry-Coulomb friction parameter** μ [-] | 0.1/0.01 steep/flat | 0.05/0.01 steep/flat | 0.1/0.01 steep/flat | 0.05/0.01 steep/flat |
| **Density of erodible layer** [kg m⁻³] | 1800 | 1800 | 1800 | 1800 |
| **Depth of erodible layer** [m] | 5 | 5 | 5 | 5 |

The volumes for Uchitel and Teztor lakes are based on values of historical events for the medium scenario and on future lake evolution projections for the large scenario. The volume of lake Uchitel varied between

30'000 and 85'000 m³ from 2010 to 2023 (MES, 2023; Zaginaev, Falatkova, et al., 2019). Reconstruction of historical events based maximum discharge values (e.g. 925 m³s⁻¹) and the size of the debris fan, however, suggest larger outburst volumes coming down the Aksay stream (Erokhin & Dikikh, 2003; Shatravin, 1978; Zaginaev et al., 2016). The medium scenario was therefore set to 200'000 m³ for lake Uchitel. Past observations of lake Teztor indicate historical lake volumes of 30'000-150'000 m³ (Erokhin

& Dikikh, 2003; Erokhin & Zaginaev, 2020), with the most recent event in 2012 releasing a volume of 74'000 m³ (Erokhin et al., 2018). While lake Uchitel forms at the same position every year, the Teztor lake complex allows for a varying position of the main lake allowing for a wider range of lake volumes.



We therefore assumed a lake volume of 100'000 m³ for the medium scenario. The large volume scenario is a worst-case scenario under realistic near future conditions, for which we assume continuing glacier
retreat as well as an expansion and coalescence of thermokarst depressions, offering more space for the two lake complexes to grow bigger. Projecting further retreat of the Aksay glacier tongue by another 50-200m, the Uchitel Lake area could increase to ~30'000-50'000 m² (compared to the 11'700 m² in 2023 (MES, 2023)). Simulations were run for an outburst volume of 300'000 m³ from lake Uchitel. For lake Teztor, we assumed a lake volume of 400'000 m³, based on possible further glacier retreat of around 200
m and the collapse and melt of buried ice, allowing for a lake area of ~ 40'000 m² (compared to 11'500 m² in 2012, (Erokhin & Zaginaev, 2020)). For all lake scenarios we simulated an outburst of the complete water volume.

Assumptions about realistic peak discharge values for the corresponding scenarios were made considering both the averaged results of volume based empirical equations proposed by Huggel et al. (2004) and by
Popov (1991), as well as estimates of historical events. Historical peak discharge estimates for lakes Uchitel and Aksay range between 300-900 m³ s⁻¹ (Erokhin et al., 2020; Erokhin & Dikikh, 2003). The 2012 peak discharge for lake Teztor was 300 m³ s⁻¹ (Erokhin et al., 2018). Both lakes could drain through underground channels due to subsurface melt, or through surface drainage due to overflow caused by a glacier calving event into the lake or by to rapid ice melt and filling of the depression. As peak discharge
has a high impact on mass flow travel time, high maximum discharge values were chosen in favour of safety.

In the Ala-Archa valley, we distinguished two areas of different flow types: a rather granular, debris dominated flow in the steep upper part or the catchment, and a more viscous and water dominated flow in the flat lower part of the catchment (cf. Fig. 3 for flat and steep areas). Flow densities were set between
1100 kg m⁻³ (hyper-concentrated flow), 1400 kg m⁻³ (viscous debris flow) and 1800 kg m⁻³ (granular debris flow). The model's frictional parameters were set based on commonly used values in the literature (Bartelt et al., 2022; Christen et al., 2010; Frank et al., 2015; Frey et al., 2018; Iribarren Anacona et al., 2018; Schneider et al., 2014). While the velocity-dependent turbulent friction parameter ξ was set to 400 m s⁻² for all sections and scenarios, values for the velocity-independent Coulomb parameter μ were
adjusted between 0.01 and 0.1 according to the scenario and the slope inclination. Erosion in RAMMS is controlled by the shear stress and the erosion rate (Bartelt et al., 2022). We defined the steep upstream part as erodible area for which we assumed a maximum erosion depth of 5 m and an erosion density of 1800 kg m⁻³, which is a common value for morainic and landslide deposits (Gan et al., 2018; Liu et al., 2020). The simulations were run on an open access digital elevation model (DEM) from ALOS PALSAR
with a 12.5 m pixel resolution. The DEM was resampled to 3 m resolution in critical areas of focus and manually adjusted to accurately represent the deflection dam and the retention basin and dam for the two structural GLOF DRM measures. The GLOFs were simulated over a distance of roughly 30 km.

Hazard maps were compiled according to a reduced 2x2 hazard matrix adapted from the methodology described in Lateltin et al. (2005) and GAPHAZ (2017) (Fig. 6). We considered only flow height for the
discrimination of high and medium intensity, as it can be expected that flow heights ≥ 0.1 m cause damage irrespective of their flow velocity. This procedure was followed to create hazard maps for the baseline case with no GLOF DRM measure in place, and repeated for the three different DRM measures. The different hazard maps were then compared to each other in terms of change in hazardous area (i.e. hazardous areas vs. non-hazardous area) and in hazard class area (area of positive vs. area of negative
hazard level change). Additionally, a weighted metric, termed 'hazard reduction score,' was computed by multiplying the level of hazard change by its pixel amount and summing the results over the entire affected area (Table 2; i.e., change in hazard class from high to low is more significant than from high to moderate). This leads to a unitless hazard reduction score, that indicates a positive change the higher the number is and indicates a total negative change if the score is below zero. This score gives a comparable indication
on the benefit of the DRM measure.





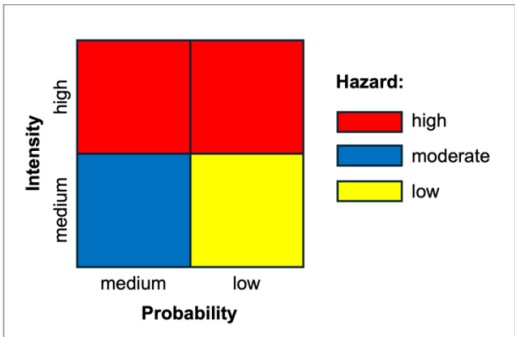

**Figure 6**: 2x2 Hazard matrix for GLOFs with hazard classes as a function of intensity and probability. The hazard classes for the medium and the large scenarios are highlighted in the shaded gray boxes. Adapted from Lateltin et al. (2005).

**Table 2**: Hazard reduction score matrix for the quantification of changes in hazard level. The higher the score, the larger is the hazard reduction. E.g., the score of a change from high to low hazard is 2. The score of a change from no hazard to low hazard is -1.

| Hazard class in baseline case subtracted by [-] Hazard class in DRM measure case | High hazard (3) | Moderate hazard (2) | Low hazard (1) | No hazard (0) |
|---|---|---|---|---|
| High hazard (3) | 0 | -1 | -2 | -3 |
| Moderate hazard (2) | 1 | 0 | -1 | -2 |
| Low hazard (1) | 2 | 1 | 0 | -1 |
| No hazard (0) | 3 | 2 | 1 | 0 |

We analyzed hazard and exposure for the valley bottom of the Ala-Archa catchment. In order to properly compare the different GLOF DRM measures, we focused on their effect in the overall analysed area in the valley bottom, as well as in different subsections, namely the Aksay fan that is mostly used for tourist activity, the settlement area of Nauka, Kashka Suu and Baytik, and the settlement area below the retention basin at the entrance of Kashka Suu (Fig. 7).





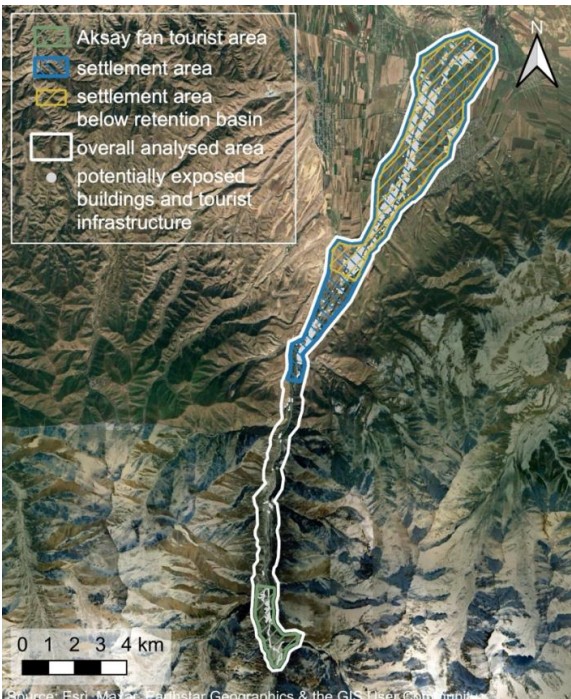

**Figure 7**: Ala-Archa catchment with the different analysis sections that were used for the hazard and exposure assessments.

### 3.3 Exposure assessment

In order to assess exposure, the GLOF hazard mapping results were complemented with information on exposed assets and people. For residential areas, the assessment was based on buildings, whereas for the National Park area it was based on the buffer zones around the most visited touristic sites. The exposure of roads, bridges and other infrastructure was not considered specifically, as it is not expected to change significantly between the different DRM measures, i.e., due to their proximity to the flow path, most bridges can be expected to be affected by mass flows irrespective of the simulated DRM measure. Roads and similar infrastructure were not considered, as we focused on the higher monetary value assets of buildings.

The mapping of buildings was done based on the regional OpenStreetMap (OSM) data set and complemented by manual mapping on the basis of aerial imagery from the web mapping services Google and Bing Maps (most recent imagery from May 2024). Buildings were treated as points and intersected with the hazard maps, which allowed to infer the number of affected buildings in each hazard class for the different DRM measures. In a survey of 763 buildings, in the Kashka Suu, Baytik, Tash-Dobo, Birbulak, Zarechnoye and Chon-Aryk villages, conducted in 2023 by experts of the Central-Asian Institute for Applied Geosciences (CAIAG), information on building type, use, occupation and condition was collected, together with information on the residents including number of inhabitants, gender, age and disabilities, beside others. The survey representatively covered about 23 % of the total amount of ca. 5500 buildings mapped in the main villages of Nauka, Kashka Suu, Baytik and Zarechnoe. The number of exposed people was extrapolated based on the average number of people per building from the representative building survey.

The mapping of tourist areas was conducted based on information collected during field visits in 2021, 2022, 2023 and 2024 and on satellite imagery (Google Earth, 2024). The tourist exposure area includes a 2 m buffer around the main touristic road and walking paths. The roads indicated by OSM were compared and adjusted with the most frequented paths extracted from Strava heat maps for hiking, running and walking activities (Strava, 2024). Tourist areas also include the main parking lots, the most popular resting and picnic places, the main campsite, and the areas of and around tourist facilities like bathrooms, restaurants, shops, the Alplager hotel, a museum, etc. Due to the large variability in tourist presence




depending on the season, weather, weekday and daytime, we compared those 'tourist zones' in terms of area rather than in terms of tourist numbers. The area size can be considered conceptually representative of the time spent in it (i.e. small area of walking paths, large area of campsite, picnic spots or parking lot).

### 3.4 Cost-benefit considerations

The benefit of GLOF DRM measures was assessed in terms of the damage avoided by the measure compared to the potential damage in the baseline simulation and compared to the estimated implementation and maintenance costs of the measure. Building damage is approximated based on expected cost of building reconstruction as reported in the literature (e.g., (Scaini et al., 2024)) and by local experts. Risk to individuals can be monetized in order to compare it to property risk in financial terms. Valuation of life for cost-benefit analyses is often referred to as Value of Statistical Life (VoSL), which is to be understood as the population's willingness to pay for marginal reductions in mortality risk. However, such monetization is sensitive as it involves value judgments about the worth of a life, and the application of VoSL is complex and varies across different cultural, economic and social contexts (e.g., EconoMe (Bundesamt für Umwelt BAFU, 2024)). Therefore, we here refrain from approximating loss of life in terms of financial cost. Cost ranges for the implementation of different GLOF DRM measures are based on the one hand, on data found in the literature for similar contexts such as Kazakhstan (e.g., (Kassenov, 2022)), and on local expert knowledge. On the other hand, we use the previously mentioned avoided damage cost as upper bound benchmark for the maximum allowable cost of a DRM measure (Bundesamt für Umwelt BAFU, 2024).

## 4 Results

### 4.1 Hazard assessment

Hazard maps were computed for the baseline case as well as for the three GLOF DRM measures. In the baseline case (Fig. 8), areas of high hazard are located along the whole valley in proximity to the river. Especially the upper half of the catchment is found to be high hazard area together with a smaller proportion of moderate hazard and an even smaller proportion of low hazard area. Results show low hazard primarily for areas below the existing dam and retention basin. Table 3 summarizes the hazard class areas and area changes for the different DRM cases in each of the analysed sections (Fig. 7).

**Table 3:** Hazard assessment for the different DRM measures and analysed sections compared to the baseline case with no implemented measure.

| | Overall analysed area | | | | Aksay fan tourist area | | | | Settlement area | | | | Settlement area below retention basin | | | |
|---|---|---|---|---|---|---|---|---|---|---|---|---|---|---|---|---|
| | baseline | drainage | deflection | retention | baseline | drainage | deflection | retention | baseline | drainage | deflection | retention | baseline | drainage | deflection | retention |
| | *hazard class area [km²]* | | | | | | | | | | | | | | | |
| all hazard classes | 5.24 | 4.07 | 4.97 | 3.38 | 1.09 | 1.04 | 0.82 | 1.09 | 2.78 | 1.77 | 2.76 | 0.90 | 1.88 | 0.99 | 1.88 | 0.02 |
| high hazard | 2.61 | 1.76 | 2.43 | 2.12 | 0.67 | 0.53 | 0.49 | 0.67 | 1.00 | 0.46 | 1.00 | 0.50 | 0.52 | 0.13 | 0.52 | 0.01 |
| moderate hazard | 1.46 | 1.75 | 1.11 | 0.96 | 0.37 | 0.47 | 0.24 | 0.37 | 0.79 | 0.91 | 0.57 | 0.29 | 0.48 | 0.54 | 0.26 | 0.00 |
| low hazard | 1.17 | 0.55 | 1.43 | 0.29 | 0.06 | 0.04 | 0.10 | 0.06 | 1.00 | 0.39 | 1.19 | 0.10 | 0.88 | 0.32 | 1.10 | 0.01 |
| | *hazard class area change compared to the baseline case with no implemented DRM measure [km²]* | | | | | | | | | | | | | | | |
| all hazard classes | - | -1.17 | -0.27 | -1.86 | - | -0.06 | -0.27 | - | - | -1.01 | -0.02 | -1.88 | - | -0.90 | 0.00 | -1.86 |
| high hazard | - | -0.86 | -0.18 | -0.49 | - | -0.14 | -0.18 | - | - | -0.54 | 0.00 | -0.49 | - | -0.39 | 0.00 | -0.51 |
| moderate hazard | - | 0.30 | -0.35 | -0.50 | - | 0.10 | -0.13 | - | - | 0.13 | -0.22 | -0.50 | - | 0.05 | -0.22 | -0.48 |
| low hazard | - | -0.61 | 0.26 | -0.87 | - | -0.02 | 0.04 | - | - | -0.60 | 0.20 | -0.90 | - | -0.56 | 0.22 | -0.87 |
| | *hazard class area change compared to the baseline case with no implemented DRM measure [%]* | | | | | | | | | | | | | | | |
| all hazard classes | - | -22 | -5 | -36 | - | -5 | -25 | - | - | -36 | -1 | -68 | - | -48 | 0 | -99 |
| high hazard | - | -33 | -7 | -19 | - | -20 | -27 | - | - | -54 | 0 | -49 | - | -76 | 0 | -99 |
| moderate hazard | - | 20 | -24 | -34 | - | 27 | -36 | - | - | 16 | -28 | -63 | - | 11 | -46 | -100 |
| low hazard | - | -53 | 23 | -75 | - | -34 | 7 | - | - | -60 | 20 | -90 | - | -63 | 25 | -98 |
| | *hazard change compared to the baseline case with no implemented DRM measure* | | | | | | | | | | | | | | | |
| increasing hazard [km²] | - | 0.02 | 0.17 | 0.03 | - | 0.00 | 0.16 | - | - | 0.00 | 0.00 | 0.03 | - | 0.00 | 0.00 | 0.00 |
| increasing hazard % | - | 0.4 | 3 | 1 | - | 0.1 | 15 | - | - | 0.1 | 0.1 | 1 | - | 0 | 0.1 | 0 |
| decreasing hazard [km²] | - | 2.47 | 0.87 | 1.88 | - | 0.23 | 0.64 | - | - | 1.83 | 0.22 | 1.88 | - | 1.53 | 0.22 | 1.88 |





| | | | | | | | | | | | | | | | |
|---|---|---|---|---|---|---|---|---|---|---|---|---|---|---|---|
| decreasing hazard % | - | 47 | 17 | 36 | - | 21 | 58 | - | - | 66 | 8 | 68 | - | 81 | 12 | 100 |
| hazard reduction score* [M] | - | 2.58 | 0.98 | 3.34 | - | 0.23 | 0.77 | - | - | 1.93 | 0.22 | 3.34 | - | 1.63 | 0.22 | 3.36 |

\* hazard reduction score = multiplication of hazard level change (Table 2) by number of pixels, summed up over the total affected area. Positive values indicate positive change (i.e. a reduction of hazard) while negative values indicate negative change (i.e. an increase of hazard). A map of the different areas analysed is shown in Fig. 7.

The total hazardous area with no implemented GLOF DRM measure corresponds to 5.24 km$^2$ in the overall analysed basin, 1.09 km$^2$ in the Aksay fan tourist area, 2.78 km$^2$ in the settlement area, and 1.88 km$^2$ when only considering the settlement area below the retention basin. Through partial lake drainage, these areas are reduced by 22 % in the overall analysed area, by 5 % in the Aksay fan tourist area, by 36 % in the settlement area and by 48 % in the settlement area below the retention basin. Through the deflection dam they are reduced by 5 % in the overall area and by 25 % in the Aksay fan tourist area, whereas they show little to no change in the settlement areas (-1 % and 0 %). Through the retention basin the hazardous area is reduced by 36 % in the overall analysed area and by 68 % and 99 % in the settlement areas, whereas there is no change for the Aksay fan tourist area, that is located upstream of this DRM measure.

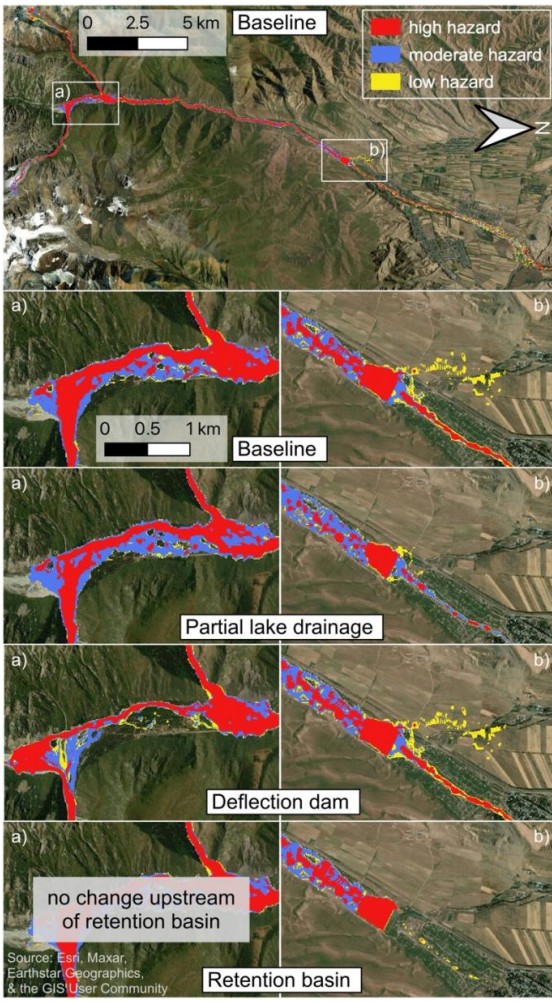

**Figure 8**: GLOF hazard maps computed for the baseline case with no implemented GLOF DRM measure (top) and close ups of a) the Aksay fan tourist area and b) the settlement area around the retention basin for the baseline case, partial lake drainage, the deflection dam, and the retention basin.

Disaggregated by hazard class (Fig. 9), partial lake drainage reduces the high hazard area by 20-76 % depending on the analysed section, and the low hazard area by 34-63 %. However, it increases the




moderate hazard area by 11-27 % depending on the analysed section. In terms of pixel level hazard change, there is an improvement in 21-81 %, and a worsening in 0-0.4 % of the hazardous area for partial lake drainage. This leads to a hazard reduction score of 0.23-2.58 depending on the analysed section.

The deflection dam reduces the high hazard area by 7 % for the overall analysed area and by 27 % for the Aksay fan tourist area but causes no reduction in the settlement areas. It reduces the moderate hazard area by 24-46 % but increases the low hazard area by 20-72 % depending on the analysed sections. In terms of pixel level hazard change, there is an improvement in 8-58 %, and a worsening in 0-15 % of the hazardous area for the deflection dam. This leads to a hazard reduction score of 0.22-0.98 depending on the analysed area section.

The retention basin reduces the high hazard area by 19-99 % for all analysed area sections except for the Aksay fan tourist area that is located higher up then the retention basin, and hence, sees no effect from this measure. Similarly, it reduces the moderate hazard area by 34-100 % and the low hazard aera by 75-98 % for all analysed sections except for the Aksay fan tourist area. In terms of pixel level hazard change, there is an improvement in 0-100 %, and a worsening in 0-1 % of the hazardous area for the retention basin. This leads to a hazard reduction score of 3.34-3.36 depending on the analysed area section, except for the Aksay fan tourist area where the score is 0.

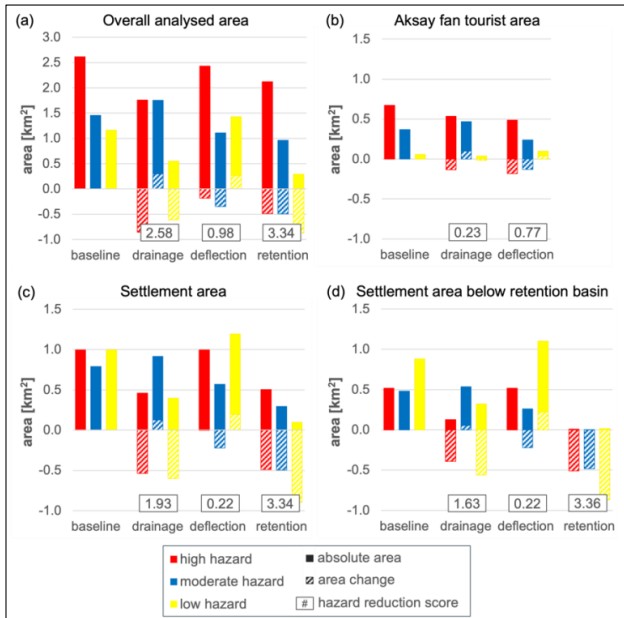

**Figure 9:** Affected area (i.e. area with max. flow height of ≥ 0.1m) distinguished by hazard class for the baseline case and the different DRM measures over the overall analysed area of the Ala-Archa valley **(a)**, over the Aksay fan **(b)**, over the settlement area **(c)**, and over the settlement area below the retention basin **(d)**. Bars with solid fill show the actual hazard areas for the different DRM measures, whereas bars with hatched fill show the difference from the baseline case. This difference refers to the change in the affected area for each DRM measure.

## 4.2 Exposure assessment

A total of 5473 buildings and the main tourist areas were mapped inside the overall analysed basin area of the Ala-Archa valley. A survey of 1287 of those buildings suggests that 75 % of them are permanently occupied whereas 25 % of them are seasonally occupied. Such seasonally used buildings are mostly vacation homes, summer houses (locally called 'dacha') and temporary buildings like yurts and picnic platforms, and a to a smaller proportion sheds, bathhouses, shops, restaurants, hotels, administrative buildings and similar. For the exposure assessment no distinction was made between permanent and temporary buildings, as GLOFs are expected to occur predominantly in summer when all the temporary buildings are in use. Table 4 summarizes the building exposure and exposure changes for the different DRM cases in each of the analysed sections (cf. Fig. 7).




**Table 4:** Exposure assessment for the different DRM measures and analysed sections compared to the baseline case with no implemented measure.

| | Overall analysed area | | | | Aksay fan tourist area | | | | Settlement area | | | | Settlement area below retention basin | | | |
|---|---|---|---|---|---|---|---|---|---|---|---|---|---|---|---|---|
| | baseline | Drainage | deflection | retention | baseline | drainage | deflection | retention | baseline | drainage | deflection | retention | baseline | drainage | deflection | retention |
| | exposure [number of buildings] | | | | | | | | | | | | | | | |
| all hazard classes | 1035 | 513 | 989 | 199 | 59 | 55 | 16 | 59 | 940 | 424 | 937 | 104 | 837 | 355 | 834 | 1 |
| high hazard | 180 | 58 | 178 | 63 | 15 | 3 | 9 | 15 | 142 | 40 | 146 | 25 | 117 | 21 | 121 | 0 |
| moderate hazard | 332 | 316 | 201 | 92 | 40 | 49 | 1 | 40 | 281 | 257 | 189 | 41 | 241 | 218 | 148 | 0 |
| low hazard | 523 | 139 | 610 | 44 | 4 | 3 | 6 | 4 | 517 | 127 | 602 | 38 | 479 | 116 | 565 | 1 |
| | exposure change compared to the baseline case with no implemented DRM measure [number of buildings] | | | | | | | | | | | | | | | |
| all hazard classes | - | -522 | -46 | -836 | - | -4 | -43 | - | - | -516 | -3 | -836 | - | -482 | -3 | -836 |
| high hazard | - | -122 | -2 | -117 | - | -12 | -6 | - | - | -102 | 4 | -117 | - | -96 | 4 | -117 |
| moderate hazard | - | -16 | -131 | -240 | - | 9 | -39 | - | - | -24 | -92 | -240 | - | -23 | -93 | -241 |
| low hazard | - | -384 | 87 | -479 | - | -1 | 2 | - | - | -390 | 85 | -479 | - | -363 | 86 | -478 |
| | exposure change compared to the baseline case with no implemented DRM measure [%] | | | | | | | | | | | | | | | |
| all hazard classes | - | -50 | -4 | -81 | - | -7 | -73 | - | - | -55 | -0.3 | -89 | - | -58 | -0.4 | -100 |
| high hazard | - | -68 | -1 | -65 | - | -80 | -40 | - | - | -72 | 3 | -82 | - | -82 | 3 | -100 |
| moderate hazard | - | -5 | -40 | -72 | - | 23 | -98 | - | - | -9 | -33 | -85 | - | -10 | -39 | -100 |
| low hazard | - | -73 | 17 | -92 | - | -25 | 50 | - | - | -75 | 16 | -93 | - | -76 | 18 | -100 |

In the baseline case with no DRM measure implemented, 1035 buildings are exposed to GLOF hazard in the overall analysed area. The overall building exposure is reduced by 50 % (522 buildings) through partial lake drainage, by 4 % (46 buildings) through the deflection dam and by 81 % (836 buildings) through the
retention basin. Although locally the percentual reduction is significantly larger, for example for the deflection dam (-73 % on the Aksay fan) and for the retention bas (-89-100 % in the settlement areas). Disaggregated by hazard class (Fig. 10), partial lake drainage reduces the exposure to high hazard by 68-82 % and to low hazard by 25-76 % depending on the analysed section. It reduces the exposure to moderate hazard by 5-10 % for all the analysed sections except for the Aksay fan tourist area, where it increases the
exposure by 23 %. This is largely because areas that were high hazard areas, classify as moderate hazard areas after lake drainage. The deflection dam reduces the exposure to high hazard by 1 % for the overall analysed area and by 40 % in the Aksay fan tourist area. However, it increases it by both 3 % in the settlement areas. It reduces exposure to moderate hazard by 33-98 % but increases exposure to low hazard by 16-50 % for the analysed sections. And the retention basin reduces the exposure to high hazard by 40-
100 %, exposure to moderate hazard by 72-100 %, and exposure to low hazard by 92-100 % for all analysed sections except for the Aksay fan tourist area.





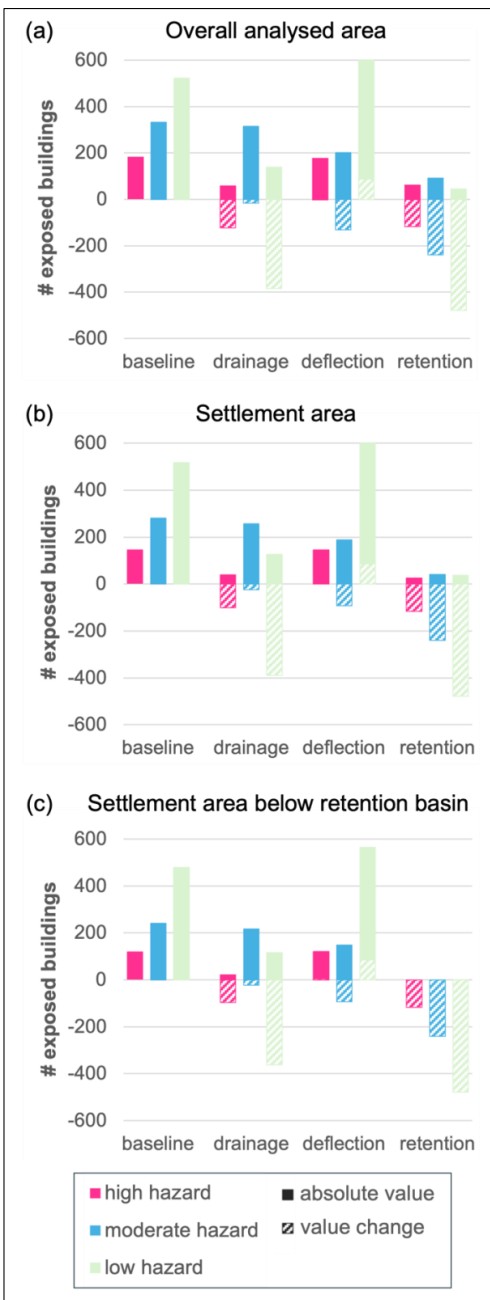

**Figure 10**: Number of exposed buildings in each hazard class area (solid filling) and the change in exposed buildings per hazard class (hatched filling) with different DRM measures compared to the baseline case of no implemented measure, for the overall analysed area **(a)**, the settlement area **(b)**, and the area of the settlements below the retention basin **(c)**.

Based on the local building exposure survey data, the average resident number per building is 4 people. For the baseline case that leads to an exposure of 4140 residents over the overall analysed basin. Based on building numbers, 90 % (3760) of those exposed residents are located in the settlement areas in the lower reaches of the analysed basin. With the enhanced retention basin, 3344 fewer residents (or 81 % less) are exposed to GLOF hazard. For partial lake drainage, it is 2088 fewer residents (or 50 % less) that are exposed, and for the deflection dam it is 184 fewer residents (or 4 % less).

In the Aksay fan tourist area, people number per building can be expected to differ from the data that was collected for the residential areas. Most of the buildings in the tourist area of the Aksay fan can be expected





to be non-residential buildings such as hotels (e.g. Alplager), museums, restaurants, shops, yurts, conference buildings, etc. For the Aksay fan tourist area, exposure of people is therefore based on information about National Park visitor numbers. The National Park Ala-Archa is visited by approximately 7'000 visitors on average on a high season weekend day according to the National Park's director (Radio Azattyk, 2024). As tourist presence largely varies and visitor movement is very dynamic,

we used the touristic area as proxy for tourist exposure. On the Aksay fan, there is a total area of roughly 75'000 m$^2$ that is mostly frequented by tourists, i.e., walking trails, parking areas and picnic areas. 57'643 m$^2$ (77 %) of that area are exposed to GLOF hazard, 20'950 m$^2$ of which are exposed to high GLOF hazard. Partial lake drainage would reduce the total exposed tourist area by 9 % and the deflection dam would reduce it by 51 %. The retention basin causes no exposure reduction for the tourist areas, as it is

located further downstream in the basin. Disaggregated by hazard class, the exposure reduction for the partial lake drainage is especially large for the high hazard area (-29 %) and a bit smaller for the low hazard area (-6 %). However, it creates an increase in the exposure of moderate hazard areas (+4 %). The deflection dam on the other hand causes the largest exposure reduction for moderate hazard (-79 %), followed by low hazard (-48 %) and high hazard (-9 %).


## 4.3 Cost-benefit evaluations

According to a regional study by Scaini et al. (2024), building reconstruction costs in Kyrgyzstan range between 175-400 USD m$^{-2}$ for different building types (e.g., unreinforced masonry, confined masonry, reinforced masonry, and reinforced concrete). However, local experts suggest significantly higher

reconstruction costs for the above average value buildings in Bishkek's surroundings of the comparatively wealthy North of the country. For instance, 40 % of the buildings assessed in the survey were built after 2000 and only 38 % of the buildings were built before 1990. Based on local conditions and experience, reconstruction costs of 600-1000 USD m$^{-2}$ with an average of 800 USD m$^{-2}$ are assumed for damage evaluation. Typical floor areas for single household 1-2 story buildings are estimated to range between

150-250 m$^2$ for Central Asian countries (Scaini et al., 2024). Based on the local exposure survey, field visits and satellite imagery, the average building has between one (71 %) and two floors (25 %) and a slightly lower estimated total area, with especially dachas being smaller, having a standard footprint of roughly 50 m$^2$ and a second floor in most cases. Assuming more heterogeneous floor areas of 100-250 m$^2$ and an average cost of 800 USD m$^{-2}$, this amounts to expected reconstruction costs of 140'000 ± 60'000

USD per building. We use the same reconstruction cost for all hazard classes (high, moderate, low), because we assume that reconstruction or at least significant rehabilitation is necessary also for flow heights between 0.1 and 1 m, especially considering the high sediment content of any expected flow. In the baseline case with 1035 GLOF exposed buildings this translates into potential damage costs of 144.9 M ± 62.1 M USD in case of a complete reconstruction of all buildings. Partial lake drainage could reduce

these costs by 73.1 M ± 31.3 M USD, the deflection dam by 6.4 M ± 2.7 M USD and the retention basin by 117 M ± 50.2 M USD.
The feasibility and cost of draining glacier lakes largely depends on the accessibility and characteristics of the lakes and purpose of the DRM measure. In Kyrgyzstan's neighbouring country Kazakhstan, annual costs for repeated drainage amount to 10'000-60'000 USD per lake (Kassenov, 2022). Based on the MES

of the Kyrgyz Republic, costs are in the range of around 4000 USD (considering transport and material cost to a close-by lake) and 6500 USD (cost of one hour of helicopter transport flight) for lake drainage. With additional equipment and labour cost, we assume local annual costs of 10'000-15'000 USD for the drainage of one lake, which results in a cost 0.5 ± 0.1 M USD for annual lake lowering of two lakes during 20 years. According to the Department of Capital Construction for Prevention and Elimination of

Emergency Situations of the MES, the construction of a protective dam (2-3 m height, including gabion net and local rocks) ranges between 1100 and 1400 USD per linear meter depending on the type of dam and local conditions. For a 500 m long and 8 m high deflection dam this results in a cost 3300-4200 USD per linear meter and a total cost of 1.9 M ± 0.2 M USD for the enhanced deflection dam. The cost of a retention basin with a capacity of 1.37 ×10$^6$ m$^3$ and a 10 m high dam is difficult to assess, especially as

such structures generally come with additional costs for feasibility studies as well as operational and maintenance costs. A cost range of 4400-5600 USD per linear meter (for a 10 m high dam) results in 1.8 ± 0.2 M USD for a 350 m dam in front of the retention basin. Based on experiences from Kazakhstan





(pers. communication, experts from Kaszelezashita; 1.1 M USD for design and construction of a simple basin) and estimates from the RESILAND CA+ project by the World Bank (10-15 M for a very
sophisticated structure with additional levee construction, regreening, etc.), costs around $2.9 \pm 0.2$ M USD can be expected for a relatively simple retention basin with a dam in the village of Kashka Suu.

Table 5 summarizes the costs and benefits of the analysed GLOF DRM measures. The range of potential building damage cost reduction of $6.4$ M $\pm 2.7$ M USD for the deflection dam is relatively low, compared to the potential damage cost reduction of partial lake drainage that is more than one order of magnitude
higher, and that of the enhanced retention basin, which is about 18 times higher. When comparing the measures' potential building damage cost reduction to their implementation cost, this results in a ratio of 146.2 for partial lake drainage over 20 years, 41.1 for the enhanced retention basin and 3.4 for the enhanced deflection dam. Hazard area reduction costs are $0.4 \pm 0.1$ USD per square meter for partial lake drainage, $1.5 \pm 0.1$ USD per square meter for the retention basin, and $7.0 \pm 0.7$ USD per square meter for
the deflection dam (Table 5). Considering the potential reduction of exposed residents, it would mean a per capita investment cost of $10'326 \pm 1'087$ USD per resident on the Aksay fan for the enhanced deflection dam, $240 \pm 48$ USD per resident for the partial lake drainage and $852 \pm 63$ USD per resident for the enhanced retention basin. However, if considering also the reduction in tourist exposure from the enhanced deflection dam and partial lake drainage, it lowers the per capita investment cost of those
measures to $657 \pm 69$ USD and $195 \pm 39$ USD per person (including residents and tourists) respectively.

**Table 5**: Benefit and cost considerations for the different GLOF DRM measures.

|  | lake drainage | deflection dam | retention basin |
|---|---|---|---|
| cost of measure [M USD] | $0.5 \pm 0.1$ | $1.9 \pm 0.2$ | $2.9 \pm 0.2$ |
| avoided building damage cost [M USD] | $73.1 \pm 31.3$ | $6.4 \pm 2.7$ | $117.0 \pm 50.2$ |
| benefit to cost ratio [-] | 146.2 | 3.4 | 41.1 |
| resident exposure reduction [#] | 2088 | 184 | 3344 |
| peak season tourist exposure reduction [#] | 473 | 2707 | - |
| investment cost per exposed resident [USD] | $239.5 \pm 47.9$ | $10326.1 \pm 1087.0$ | $852.3 \pm 62.8$ |
| investment cost per exposed resident + tourist [USD] | $195.2 \pm 39.1$ | $657.4 \pm 69.2$ | - |
| tourist area exposure reduction [m$^2$] | 5130 | 29255 | - |
| cost per m$^2$ reduction of exposed tourist area [USD] | $97.5 \pm 19.5$ | $65.0 \pm 6.8$ | - |
| reduction of tourist exposure area per USD [m$^2$] | 0.01 | 0.02 | - |
| hazard area reduction [km$^2$] | 1.17 | 0.27 | 1.86 |
| cost per m$^2$ reduction of hazard area [USD] | $0.4 \pm 0.1$ | $7.0 \pm 0.7$ | $1.5 \pm 0.1$ |
| reduction of hazard area per USD [m$^2$] | 2.34 | 0.14 | 0.65 |

## 5 Discussion

### 5.1 Hazard reduction

Depending on the DRM strategy and the goals of the DRM measures, different measures should be favored in decision-making processes. For example, the retention basin is preferable, if the interest lies in the reduction of hazardous area or in the weighted reduction of the hazard level in the overall and settlement areas. Effective hazard reduction on the Aksay fan tourist area, rather calls for a deflection dam. Partial lake drainage is the second most effective option in all analysed sections, considering both
hazard area and hazard level reduction (with the second highest hazard reduction score in each situation). In that sense it is a more flexible measure than the deflection dam, that has a large effect on the Aksay fan, but not further downstream, or the retention basin, that has a large effect on the downstream areas, but none on the upstream areas. Here, the hazard reduction score serves as metric indicating the overall effectiveness of a DRM measure. As it is positive for all three DRM measures in all analysed sections,
the implementation of either of the three measures can be expected to be beneficial from a GLOF hazard reduction point of view.

For partial lake drainage, we conceptually only consider the effect of the lowered lake volume in the simulation of the mass movement. However, possibly, partial lake drainage could have an impact not only on the intensity and magnitude of the GLOF, but also on the probability of an outburst. Hence, partial lake
lowering could lower the probability of a GLOF initiating but uncertainties related to the exact physical processes of the lake drainage prevents any quantification, and even the sign of change in probability of occurrence is not certain. In the case of sustained lake lowering permanently reducing outburst probability,



the measure of partial lake drainage would outperform the other two measures both in terms of reduced hazard area and reduced exposure. Hence, by keeping probability static, we may systematically underestimate the actual benefits of partial lake drainage relative to the other two GLOF DRM measures.

## 5.2 Exposure reduction

In terms of GLOF risk, changes in GLOF hazard level and area are relevant only when spatially coinciding with exposed assets. This study specifically considers building exposure and tourist area exposure. Percentual building exposure reduction differs for each measure depending on the analysed section and hazard class and corresponds to different absolute numbers of reduced buildings (see Table 5). The reduction of buildings exposed to high hazard is especially relevant, considering that buildings exposed to high hazard (i.e., flow heights of more than 1 m) are more probable to completely collapse and cause not only building damage but also result in fatalities. While the number of exposed buildings in the Aksay fan tourist area - and therefore also the potential for building exposure reduction - are comparatively low, it features a relevant exposed tourist area. Assuming as a worst-case scenario, 7'000 tourists (Radio Azattyk, 2024) distributed homogeneously throughout the tourist area with 5335 (equivalent to 76 % exposed tourist area in the baseline case) of them being GLOF-exposed, this could translate to the protection of 2707 tourists (equivalent to -51 % tourist exposure area) for the deflection dam and of 473 tourists (equivalent to -9 % tourist exposure area) for partial lake drainage vs. no protection of tourists for the retention basin. It is, however, important to note that this number is a lot more variable than the downstream building and resident exposure, as tourist movement is very dynamic. For instance, for a GLOF occurring at night on a bad weather weekday off-season, tourist presence may be minimal, which would also reduce the tourist exposure reduction effect of the deflection dam to a minimum.

Kyrgyzstan has experienced annual population growth and progressive urbanization since 2000 (UNESCAP, 2020; World Bank, 2024). The country's capital, Bishkek, and its vicinity is particularly experiencing rapid development (UNESCAP, 2020). With roughly 40 % of the surveyed buildings having been constructed after 2000, the built environment in Ala-Archa has been rapidly growing. Due to limited available land, there is an ongoing process of transforming territories and reclassifying land with primarily, pastures and arable lands being repurposed for the construction of infrastructure and settlements. This means that exposure and damage potential are increasing and are expected to rise further in the future.

## 5.3 Vulnerability considerations

We did not systematically analyse the vulnerability component of risk in the Ala-Archa catchment. However, according to the 2023 local exposure and social survey results, physical vulnerability is very high in the riverbank zone and settlements lack permanent protective infrastructure and include many buildings and infrastructure located in areas prone to waterlogging. At the same time, there is a constant construction of new residential and recreational houses within 50 meters of the river shore. Particularly socially vulnerable are, for example, populations such as the elderly, children and disabled people, as well as residents of informal settlements, who have limited access to information, emergency services and resources (Cutter, 2024). Of the people surveyed in the exposed areas, the percentage of female and male residents is 45 % and 55 % respectively. 14 % are 60 years old or older and 26 % are 10 years old or younger, and 2 % of the residents have some sort of disability. This results in 42 % of the local population potentially having difficulties evacuating and reaching safe ground in case of a GLOF. 13 % of the households have cattle, which may additionally be exposed to GLOFs. In the social survey 35 % of respondents noted that they had previously experienced GLOFs, floods or mudflows, but there were no serious consequences and losses of assets. However, in many communities, GLOF risk awareness remains low, and preparedness is very limited. In addition, local residents often lack clear evacuation plans and emergency training. Furthermore, the evacuation of National Park visitors could prove difficult, as the only road accessing the tourist area would most likely get damaged in the event of a GLOF, as was the case during the debris flow in 2003 (Erokhin & Dikikh, 2003; Kim & Gruzdov, 2003).





### 5.4 Discussion of cost

Hazard area reduction cost per square meter and per-capita investment costs per exposed resident and per exposed tourist were calculated for the different DRM measures (see Table 5). However, it is important to be aware that this hazard reduction per spent US dollar cannot be upscaled linearly, as for some of the
area it may be more difficult to reduce the hazard.

The very high benefit to cost ratio of partial lake lowering is due to both, relatively low implementation cost and relatively high potential building damage reduction. The implementation costs are particularly low thanks to the easily accessible location of the considered lakes (proximity to the capital, relatively low elevation, easy terrain to land and move in, well predictable weather, availability of shelter and hiking
paths, etc.) which allows for minimal helicopter transport cost and efficient and safe working conditions that are required for repeated drainage efforts. As material transport is the most expensive component of lake drainage, the benefit to cost ratio can be expected to be lower in less accessible areas. It is important to note, that lake drainage through siphoning and pumping is only an effective measure if it is conducted consistently and timely whenever lake levels rise above an agreed upon threshold (Niggli et al., 2024;
Portocarrero Rodríguez, 2014; Reynolds et al., 1998). It not only comes with continuous annual implementation costs (also beyond the considered 20 years), but also with additional costs for monitoring. While permanent lake drainage is also an option, the cost for building permanent drainage channels or tunnels is significantly higher. In addition, changes in the hazard landscape such as the emergence of additional lakes can be expected to increase the costs proportionally.

In addition, it must be noted that partial lake drainage does not reduce the risks stemming from other potential mass movements like floods, debris flows or GLOFs from other sub-catchments. The enhanced deflection dam and retention basin, however, have the potential to reduce the risk not only from GLOFs originating from the two considered lakes Uchitel and Teztor, but act as multi-hazard DRM measures that are effective on any mass movement process from the area upstream of the measure. Maintenance costs,
such as sediment removal following an event, will be required for the deflection dam and retention basin as well. For the latter, these costs will arise regardless, after several years, even under normal sedimentation rates. These maintenance costs have also not been accounted for in the presented cost-benefit evaluations. Further, lake lowering does not require large constructions in recreational areas of the Ala-Archa national park or in built-up residence areas, respectively, which can be a relevant consideration
in terms of preservation of the natural scenery. Other information that are not represented in the benefit to cost ratio is the time beyond the planned lifetime of either the enhanced deflection dam or the enhanced retention basin, or additional direct or indirect costs and benefits (e.g., stemming from effects of water regulation and irrigation, aestetics, value of land, etc.).

For the assessment of the benefits (i.e., avoided damage cost), we monetarily estimate building cost only.
However, additional costs can be expected to arise through damage to other types of infrastructure, such as electricity and water infrastructure or roads and bridges, and to people. For instance, the outburst of Lake Teztor in 2012 (comparable to the medium scenario used for our assessments in terms of volume (74'000 m$^3$) and maximum discharge (340 m$^3$ s$^{-1}$) damaged water pipelines which caused losses of around 100'000 USD in the hydro energy sector (Erokhin & Zaginaev, 2020). Damage to such infrastructure may
not only cause additional direct and indirect costs, but can also hinder search and rescue, evacuation and immediate relief activities, as reported after the GLOF in 2003 (Kim & Gruzdov, 2003). The benefit to cost ratio exceeds 1 for all measures, already when only considering avoided costs of potential building damage. The ratio can be expected to be even higher, when considering additional avoided costs and intangible losses (Hugenbusch & Neumann, 2021; Menk et al., 2022).

### 5.5 Challenges and limitations

While worst-case GLOF scenarios are often most relevant for informing DRM, it is not always clear, what "realistic" worst-case scenarios are over what time horizon (Emmer et al., 2022). Firstly, this poses a challenge for the choice of the scenario and secondly, it complicates the evaluation of costs and benefits. Based on the local geomorphology, a catastrophic outburst with complete drainage over a very short time
span is rather unlikely for the lakes Uchitel and Teztor. It is more realistic to assume high discharge values due to melt and precipitation causing mixed mass flow events of higher frequency but lower magnitude. At the same time, we did not consider combined GLOF events from several lakes at the same time, as





could, for instance, be the case in a situation of prolonged heat with strong melt followed by intense rainfall over the whole catchment. Also cascading events were not specifically considered. Local experts consider that some of the worst-case scenarios may be cascading events with a blockage of the main Ala-Archa River (e.g., due to a GLOF or landslide) in the upper reaches of the main valley, and/or even the overtopping and destruction of the present or enhanced retention basin further downstream. While this would have catastrophic consequences potentially reaching far along into the city of Bishkek, it is to be assumed that a blockage of the main valley would provide enough time to secure the downstream areas, limiting human losses and potentially reducing damages. A controlled, gradual safe breaching of the blockage may also be feasible. Nevertheless, these considerations give room for some reservations against the measure of the retention basin, that could in such a case potentially act as a multiplying factor for cascading hazards, which is not the case for the other two measures.

In this regard, a combination of several GLOF DRM measures may be more appropriate. For example, while the retention basin can be expected to significantly reduce GLOF and other flood or mudflow risk in large parts of the settlement area, it will not reduce the hazard or exposure upstream, and it loses its effect in case of an overtopping of the basin. A combination with partial lake drainage to reduce the hazard, or with a deflection dam to reduce the exposure, would offer a possibility to reduce these remaining risks. Despite the resulting higher costs, such a combination could still prove beneficial, especially, considering the high potential damage costs to be averted.

GLOF risk is often characterized by low-probability but high-impact events, for which quantitative probabilities are difficult to calculate statistically in a meaningful way (especially for non-reoccurring GLOFs). However, without consideration of the probability of a GLOF, the cost-benefit analysis is deterministic, meaning that the benefits have a probability of 100 % and, consequently, are largely overestimated (Mechler, 2016). Because we neglect quantitative GLOF probability, our cost-benefit considerations are useful for comparison between the different DRM measures, but are less valid in absolute terms. Nonetheless, potential benefits are very high. Beside monetary costs, non-market or intangible effects, such as loss of life or health impacts, loss and damage to identity creating natural assets or ecosystem services are key considerations for DRM that are not financially quantified here. While analysis of risk to individuals is essential for quantifying the benefits of measures aimed at saving lives, established techniques that assign a monetary value to human life in any way introduce substantial controversy, as they include value judgments (WB & UN, 2010). But even without numbering the avoided costs due to these challenges, a reduction of the exposure of people (both residents and tourists) greatly improves the benefit of the measures compared to their costs. Some of these points, such as representing disaster risk, assessing intangibles, assessing combinations of measures, and the role of spatial and temporal scales are known and inherent challenges of DRM CBA (Mechler, 2016).

## 5.6 DRM strategy and decision-making

Several other considerations and factors should be made and taken into account for appropriate decision-making.

GLOF DRM should account for and be flexible to future changes in the environment, be embedded in the wider multi-hazard management context, and consider the cascading nature of those hazards and their impacts (Niggli et al., 2024). Often this means, working with combinations of DRM measures that address different aspects of risk and timescales, rather than relying on one measure alone.

For example, with increasing exposure being one of the main drivers of GLOF risk in Ala-Archa, DRM measures should have a focus on the local exposure. Beside a potential combination of a retention basin and partial lake drainage, land-use planning and EWS are robust complementary measures limiting impacts on future planned built areas and infrastructure and reducing the risk posed to the numerous tourists visiting the valley. Institutionalized spatial planning based on hazard maps induces limited direct costs, while offering large potential benefits in terms of avoided damage. EWS hold limited benefits in terms of protecting built assets, but allow for the evacuation and protection of thousands of people at a relatively low cost, with implementation costs seen from past projects in high mountain Asia and Central Asia in the range of 0.5-1 M USD for an EWS (Ives et al., 2010; Wang et al., 2022).

When elaborating such DRM strategies, meaningful inclusion and engagement of local and informal institutions is fundamental. Any measure is only effective, if incorporated in the local cultural and socio-



economic context of values, beliefs and priorities, and if fully accepted and broadly supported by the local population and authorities (Huggel et al., 2020).

A strong institutional framework and stable governance (e.g., official guidelines and regulations, clear roles and responsibilities, collaborations among sectors and institutions, communication across scales) are essential for sustainable DRM. This means that it may be beneficial to allocate resources to bottom-up or
at least needs-oriented approaches that aim at the empowerment of the local people and stakeholders, before investing into specific GLOF DRM measures.

## 6 Conclusion

In this study we undertook a comparative analysis of the effectivity of three GLOF DRM measures in the Kyrgyz Ala-Archa catchment. Based on GLOF simulations with the numerical modelling software
RAMMS, we elaborated hazard and exposure maps for partial lake drainage, a deflection dam and a retention basin and for the case of no implemented DRM measure. We compared the results in terms of hazard and exposure change and analysed them with respect to cost and benefit. The hazard class area change and the hazard reduction score indicate that the enhanced retention basin has the largest effect on hazard reduction. The building and local resident exposure is also most reduced through the enhanced
retention basin whereas the tourist exposure is strongly reduced by the enhanced deflection dam. The benefit to cost ratio is above 1 for all three measures and highest for partial lake drainage. Nevertheless, additional aspects should be taken into account for decision-making. This includes consideration of additional costs and benefits such as intangible, indirect and maintenance costs and benefits, and of potential event probability changes due to some measures or combinations of measures. Complementary
measures aiming at additional components of risk, such as land-use planning and early warning reducing vulnerability and exposure, should be considered. It is important to systematically assess and evaluate DRM measures before implementing, in order for it to be more broadly supported and more effective. The method elaborated here, shall serve a basis for discussion and to institutionalize transparent, systematic and comparable GLOF DRM planning. While the method and considerations can be translated into similar
GLOF and debris flow settings, there is a need for future research to focus on the cascading nature of such hazards and their impacts. Data on local conditions for hazard simulation, for exposure and vulnerability assessment and for cost and benefit considerations are essential but often sparse in mountain regions. Systematic collection of such data to understand possible hazard and impact cascades, as well as the retrieval of local vulnerability, perceptions and needs are at the basis of effective GLOF DRM planning.
Authorities are encouraged to use such data and the approaches presented here to meaningfully engage with all affected communities and stakeholders to systematically evaluate and refine DRM plans.

## Acknowledgements

This study was conducted in the framework of the project 'Reducing vulnerabilities of populations in Central Asia from glacier lake outburst floods in a changing climate' (GLOFCA), an initiative funded by
the Adaptation Fund and executed by UNESCO and the University of Zurich, Switzerland.

We would like to acknowledge the valuable input and feedback on GLOF modelling provided by Alessandro Cicoira and Claudius Brüniger, and the kind support of Marc Christen with the RAMMS::Debrisflow software. Further, we thank Sergey Aleksandrovich Erokhin for introducing us to the Ala-Archa national park, its history of mass movement events and related research.

## 650   Author contributions

Laura Niggli, Holger Frey, Simon Allen and Christian Huggel contributed to the study conception and design. The exposure and vulnerability data collection was conducted by Nazgul Alybaeva under authority of Bolot Moldobekov. Additional data was collected by Laura Niggli. Data processing and analysis was performed by Laura Niggli. Relevant local expert knowledge (e.g. on cost, RAMMS input parameters,



etc.) was provided by Vitalii Zaginaev. The first draft of the manuscript was written by Laura Niggli. All authors commented on previous versions of the manuscript and read and approved the final manuscript.

**Statement of declaration**

The authors have no relevant financial or non-financial interests to disclose.

**Sources**

Allen, S., Rastner, P., Arora, M., Huggel, C., & Stoffel, M. (2016). Lake outburst and debris flow disaster at Kedarnath, June 2013: hydrometeorological triggering and topographic predisposition. *Landslides*, *13*(6), 1479–1491. https://doi.org/10.1007/s10346-015-0584-3

Allen, S., Zhang, G., Wang, W., Yao, T., & Bolch, T. (2019). Potentially dangerous glacial
lakes across the Tibetan Plateau revealed using a large-scale automated assessment approach. *Science Bulletin*, *64*(7), 435–445. https://doi.org/10.1016/j.scib.2019.03.011

Ballesteros Cánovas, J. A., Stoffel, M., Corona, C., Schraml, K., Gobiet, A., Tani, S., Sinabell, F., Fuchs, S., & Kaitna, R. (2016). Debris-flow risk analysis in a managed torrent based on a stochastic life-cycle performance. *Science of the Total Environment*, *557–558*, 142–
153. https://doi.org/10.1016/j.scitotenv.2016.03.036

Bartelt, P., Bieler, C., Bühler, Y., Christen, M., Deubelbeiss, Y., Graf, C., McArdell, B., Salz, M., & Schneider, M. (2022). *RAMMS: DEBRISFLOW User Manual v1.8.0*.

Benson, C., & Twigg, J. (2004). *Measuring Mitigation: Methodologies for assessing natural hazard risks and the net benefits of mitigation - a scoping study*.
www.proventionconsortium.org

Bernard, M., Boreggio, M., Degetto, M., & Gregoretti, C. (2019). Model-based approach for design and performance evaluation of works controlling stony debris flows with an application to a case study at Rovina di Cancia (Venetian Dolomites, Northeast Italy). *Science of the Total Environment*, *688*, 1373–1388.
https://doi.org/10.1016/j.scitotenv.2019.05.468

Bundesamt für Umwelt BAFU. (2024). *EconoMe 5.2 Wirkung und Wirtschaftlichkeit von Schutzmassnahmen gegen Naturgefahren*. Https://Econome.Ch. https://econome.ch/eco_work/index.php

Cardona, O. D., Ordaz, M. G., Yamin, L. E., Marulanda, M. C., & Barbat, A. H. (2008).
Earthquake loss assessment for integrated disaster risk management. *Journal of Earthquake Engineering*, *12*(SUPPL. 2), 48–59. https://doi.org/10.1080/13632460802013495

Chen, S. C., Wu, C. Y., & Huang, B. T. (2010). The efficiency of a risk reduction program for debris-flow disasters - A case study of the Songhe community in Taiwan. *Natural*
*Hazards and Earth System Science*, *10*(7), 1591–1603. https://doi.org/10.5194/nhess-10-1591-2010

Chiou, I. J., Chen, C. H., Liu, W. L., Huang, S. M., & Chang, Y. M. (2015). Methodology of disaster risk assessment for debris flows in a river basin. *Stochastic Environmental Research and Risk Assessment*, *29*(3), 775–792. https://doi.org/10.1007/s00477-014-
695 0932-1

Christen, M., Kowalski, J., & Bartelt, P. (2010). RAMMS: Numerical simulation of dense snow avalanches in three-dimensional terrain. *Cold Regions Science and Technology*, *63*(1–2), 1–14. https://doi.org/10.1016/j.coldregions.2010.04.005

Cutter, S. L. (2024). The origin and diffusion of the social vulnerability index (SoVI).
*International Journal of Disaster Risk Reduction*, *109*. https://doi.org/10.1016/j.ijdrr.2024.104576



Dan, M. B. (2018). Decision making based on benefit-costs analysis: Costs of preventive retrofit versus costs of repair after earthquake hazards. *Sustainability (Switzerland)*, *10*(5). https://doi.org/10.3390/su10051537

Emmer, A. (2024). Understanding the risk of glacial lake outburst floods in the twenty-first century. *Nature Water*, *2*(7), 608–610. https://doi.org/10.1038/s44221-024-00254-1

Emmer, A., Allen, S. K., Carey, M., Frey, H., Huggel, C., Korup, O., Mergili, M., Sattar, A., Veh, G., Chen, T. Y., Cook, S. J., Correas-Gonzalez, M., Das, S., Diaz Moreno, A., Drenkhan, F., Fischer, M., Immerzeel, W. W., Izagirre, E., Joshi, R. C., … Yde, J. C. (2022). Progress and challenges in glacial lake outburst flood research (2017-2021): a research community perspective. *Natural Hazards and Earth System Sciences*, *22*(9), 3041–3061. https://doi.org/10.5194/nhess-22-3041-2022

Erokhin, S. A., Chontoev, D. T., & Zaginaev, V. V. (2020). Outburst-prone lakes of Kyrgyzstan. In *Print Express Publishing*.

Erokhin, S. A., & Dikikh, A. N. (2003). Evaluation of the danger of debris flows and flood flows in the territory of the Ala-Archa National Park. In *Proceedings of the National Academy of Sciences of the Kyrgyz Republic* (Issue 4, pp. 130–139).

Erokhin, S. A., & Zaginaev, V. V. (2020). Types of mountain lakes in Kyrgyzstan by the level of their outburst hazard. *Georisk*, *14*(3), 78–86. https://doi.org/10.25296/1997-8669-2020-14-3-78-86

Erokhin, S. A., Zaginaev, V. V., Meleshko, A. A., Ruiz-Villanueva, V., Petrakov, D. A., Chernomorets, S. S., Viskhadzhieva, K. S., Tutubalina, O. V., & Stoffel, M. (2018). Debris flows triggered from non-stationary glacier lake outbursts: the case of the Teztor Lake complex (Northern Tian Shan, Kyrgyzstan). *Landslides*, *15*(1), 83–98. https://doi.org/10.1007/s10346-017-0862-3

Frank, F., McArdell, B. W., Huggel, C., & Vieli, A. (2015). The importance of entrainment and bulking on debris flow runout modeling: Examples from the Swiss Alps. *Natural Hazards and Earth System Sciences*, *15*(11), 2569–2583. https://doi.org/10.5194/nhess-15-2569-2015

Frey, H., Huggel, C., Chisolm, R. E., Baer, P., McArdell, B., Cochachin, A., & Portocarrero, C. (2018). Multi-Source Glacial Lake Outburst Flood Hazard Assessment and Mapping for Huaraz, Cordillera Blanca, Peru. *Frontiers in Earth Science*, *6*. https://doi.org/10.3389/feart.2018.00210

Gan, F., He, B., & Wang, T. (2018). Water and soil loss from landslide deposits as a function of gravel content in the Wenchuan earthquake area, China, revealed by artificial rainfall simulations. *PLoS ONE*, *13*(5). https://doi.org/10.1371/journal.pone.0196657

GAPHAZ. (2017). *Assessment of Glacier and Permafrost Hazards in Mountain Regions: Technical Guidance Document*.

Google Earth. (2024). *Image 2024 Airbus*. Google Earth Pro 7.3.6.9796. http://www.google.com/earth/index.html

Haeberli, W., Schaub, Y., & Huggel, C. (2017). Increasing risks related to landslides from degrading permafrost into new lakes in de-glaciating mountain ranges. *Geomorphology*, *293*, 405–417. https://doi.org/10.1016/j.geomorph.2016.02.009

Hoyos, M. C., & Silva, V. (2022). Exploring benefit cost analysis to support earthquake risk mitigation in Central America. *International Journal of Disaster Risk Reduction*, *80*. https://doi.org/10.1016/j.ijdrr.2022.103162

Hudson, P., & Wouter Botzen, W. J. (2019). Cost–benefit analysis of flood-zoning policies: A review of current practice. *Wiley Interdisciplinary Reviews: Water*, *6*(6). https://doi.org/10.1002/WAT2.1387

Hugenbusch, D., & Neumann, T. (2021). *Enhancing efficiency in humanitarian action through reducing risk. A study on cost-benefit of disaster risk reduction*. www.aktion-deutschland-hilft.de

Huggel, C., Cochachin, A., Drenkhan, F., Fluixá-Sanmartín, J., Frey, H., García Hernández, J., Jurt, C., Muñoz, R., Price, K., & Vicuña, L. (2020). Glacier Lake 513, Peru: Lessons for



755       early warning service development. *WMO Bulletin*, *69*(1), 45–52. https://library.wmo.int/doc_num.php?explnum_id=10223

Huggel, C., Haeberli, W., Kääb, A., Bieri, D., & Richardson, S. (2004). An assessment procedure for glacial hazards in the Swiss Alps. *Canadian Geotechnical Journal*, *41*(6), 1068–1083. https://doi.org/10.1139/T04-053

Immerzeel, W. W., Lutz, A. F., Andrade, M., Bahl, A., Biemans, H., Bolch, T., Hyde, S., Brumby, S., Davies, B. J., Elmore, A. C., Emmer, A., Feng, M., Fernández, A., Haritashya, U., Kargel, J. S., Koppes, M., Kraaijenbrink, P. D. A., Kulkarni, A. V., Mayewski, P. A., … Baillie, J. E. M. (2020). Importance and vulnerability of the world's water towers. *Nature*, *577*(7790), 364–369. https://doi.org/10.1038/s41586-019-1822-y

IPCC. (2018). *Special Report: Global Warming of 1.5°C. Annex 1: Glossary*.

Ives, J. D., Shrestha, R. B., & Mool, P. K. (2010). *Formation of Glacial Lakes in the Hindu Kush-Himalayas and GLOF Risk Assessment*.

Kassenov, M. (2022). *Report providing a comprehensive listing and description of GLOF and glacial mudflow mitigation and prevention methods over the past 50 years in Kazakhstan*.

Kenny, C. (2009). *Why Do People Die in Earthquakes? The Costs, Benefits and Institutions of Disaster Risk Reduction in Developing Countries*. http://econ.worldbank.org.

Kim, A., & Gruzdov, Y. (2003, July 25). Nights of fear. *MSN LLC*. https://www.msn.kg/ru/news/4699/

Kolenko, A., Teysseire, P., & Zimmermann, M. (2004). Safety concept for debris flow hazards.
The Täsch case-study. *Internationales Symposion Intrapraevent*, 193–206.

Kopp, R. J., Krupnick, A. J., & Toman, M. (1997). *Cost-Benefit Analysis and Regulatory Reform: An Assessment of the Science and the Art*.

Kull, D., Mechler, R., & Hochrainer-Stigler, S. (2013). Probabilistic cost-benefit analysis of disaster risk management in a development context. *Disasters*, *37*(3), 374–400.
https://doi.org/10.1111/disa.12002

Lateltin, O., Haemmig, C., Raetzo, H., & Bonnard, C. (2005). Landslide risk management in Switzerland. *Landslides*, *2*(4), 313–320. https://doi.org/10.1007/s10346-005-0018-8

Liu, M., Zhang, Y., Tian, S. feng, Chen, N. sheng, Mahfuzr, R., & Javed, I. (2020). Effects of loose deposits on debris flow processes in the Aizi Valley, southwest China. *Journal of*
*Mountain Science*, *17*(1), 156–172. https://doi.org/10.1007/s11629-019-5388-9

Mani, P., Allen, S., Evans, S. G., Kargel, J. S., Mergili, M., Petrakov, D., & Stoffel, M. (2023). Geomorphic Process Chains in High-Mountain Regions—A Review and Classification Approach for Natural Hazards Assessment. *Reviews of Geophysics*, *61*(4). https://doi.org/10.1029/2022RG000791

Mechler, R. (2016). Reviewing estimates of the economic efficiency of disaster risk management: opportunities and limitations of using risk-based cost–benefit analysis. *Natural Hazards*, *81*(3), 2121–2147. https://doi.org/10.1007/s11069-016-2170-y

Mechler, R., & The Risk to Resilience Study Team. (2008). *The Cost-Benefit Analysis Methodology, From Risk to Resilience Working Paper No. 1*.

Menk, L., Schinko, T., Karabaczek, V., Hagen, I., & Kienberger, S. (2022). What's at stake? A human well-being based proposal for assessing risk of loss and damage from climate change. *Front. Clim.*, *4*. https://doi.org/doi:10.3389/fclim.2022.1032886

MES. (2023). *Monitoring, forecasting of dangerous processes and phenomena on the territory of the Kyrgyz Republic (21st ed. with amendments and additions)* (Issues 21st ed. with
amendments and additions).

Niggli, L., Allen, S., Frey, H., Huggel, C., Petrakov, D., Raimbekova, Z., Reynolds, J., & Wang, W. (2024). GLOF Risk Management Experiences and Options: A Global Overview. In *Oxford Research Encyclopedia of Natural Hazard Science*. Oxford University Press. https://doi.org/10.1093/acrefore/9780199389407.013.540

Popov, N. (1991). Assessment of glacial debris flow hazard in the North Tien-Shan. *Proceedings of the Soviet-China-Japan Symposium and Field Workshop on Natural Disasters*, 384–39.





Portocarrero Rodríguez, C. A. (2014). *The Glacial Lake Handbook. Reducing risk from dangerous glacial lakes in the Cordillera Blanca, Peru*.

Radio Azattyk. (2024, September 8). Власти КР намерены ограничить въезд легкового транспорта в природный парк «Ала-Арча». *Radio Free Europe/Radio Liberty (RFE/RL), Kyrgyz Service*. https://rus.azattyk.org/a/33111563.html

Rai, R. K., van den Homberg, M. J. C., Ghimire, G. P., & McQuistan, C. (2020). Cost-benefit analysis of flood early warning system in the Karnali River Basin of Nepal. *International*
*Journal of Disaster Risk Reduction*, *47*. https://doi.org/10.1016/j.ijdrr.2020.101534

Reynolds, J. M., Dolecki, A., & Portocarero, C. (1998). The construction of a drainage tunnel as part of glacial lake hazard mitigation at Hualcán, Cordillera Blanca, Peru. In J. G. Maund & M. Eddleston (Eds.), *Geohazards in Engineering Geology* (Geological, pp. 41–48). Engineering Geology Special Publications.

Riedel, I., & Guéguen, P. (2018). Modeling of damage-related earthquake losses in a moderate seismic-prone country and cost–benefit evaluation of retrofit investments: application to France. *Natural Hazards*, *90*, 639–662. https://doi.org/10.1007/s11069-017-3061-6

Sattar, A., Allen, S., Mergili, M., Haeberli, W., Frey, H., Kulkarni, A. V., Haritashya, U. K., Huggel, C., Goswami, A., & Ramsankaran, R. (2023). Modeling Potential Glacial Lake
Outburst Flood Process Chains and Effects From Artificial Lake-Level Lowering at Gepang Gath Lake, Indian Himalaya. *Journal of Geophysical Research: Earth Surface*, *128*(3). https://doi.org/10.1029/2022JF006826

Scaini, C., Tamaro, A., Adilkhan, B., Sarzhanov, S., Ismailov, V., Umaraliev, R., Safarov, M., Belikov, V., Karayev, J., & Faga, E. (2024). A new regionally consistent exposure
database for Central Asia: population and residential buildings. *Natural Hazards and Earth System Sciences*, *24*(3), 929–945. https://doi.org/10.5194/nhess-24-929-2024

Schneider, D., Huggel, C., Cochachin, A., Guillén, S., & García, J. (2014). Mapping hazards from glacier lake outburst floods based on modelling of process cascades at Lake 513, Carhuaz, Peru. *Advances in Geosciences*, *35*(January), 145–155.
https://doi.org/10.5194/adgeo-35-145-2014

Schwanghart, W., Worni, R., Huggel, C., Stoffel, M., & Korup, O. (2016). Uncertainty in the Himalayan energy-water nexus: Estimating regional exposure to glacial lake outburst floods. *Environmental Research Letters*, *11*(7). https://doi.org/10.1088/1748-9326/11/7/074005

Shatravin, V. I. (1978). *Report on the expert examination of the Aksay debris flow source*.

Shreve, C. M., & Kelman, I. (2014). Does mitigation save? Reviewing cost-benefit analyses of disaster risk reduction. *International Journal of Disaster Risk Reduction*, *10*(PA), 213–235. https://doi.org/10.1016/j.ijdrr.2014.08.004

Shugar, D. H., Burr, A., Haritashya, U. K., Kargel, J. S., Watson, C. S., Kennedy, M. C.,
Bevington, A. R., Betts, R. A., Harrison, S., & Strattman, K. (2020). Rapid worldwide growth of glacial lakes since 1990. *Nature Climate Change*, *10*(10), 939–945. https://doi.org/10.1038/s41558-020-0855-4

Strava. (2024). *Global Heatmap*. https://www.strava.com/maps/global-heatmap

Taylor, C., Robinson, T. R., Dunning, S., Rachel Carr, J., & Westoby, M. (2023). Glacial lake
outburst floods threaten millions globally. *Nature Communications*, *14*(1). https://doi.org/10.1038/s41467-023-36033-x

UNESCAP. (2020). *Urbanization and resource trends in Kyrgyzstan*. https://www.unescap.org/about/member-states.

Wang, W., Zhang, T., Yao, T., & An, B. (2022). Monitoring and early warning system of
Cirenmaco glacial lake in the central Himalayas. *International Journal of Disaster Risk Reduction*, *73*. https://doi.org/10.1016/j.ijdrr.2022.102914

WB, & UN. (2010). *Natural hazards, unnatural disasters. The economics of effective prevention*. https://doi.org/10.1596/978-0-8213-8050-5

Willenbockel, D. (2011). *A Cost-Benefit Analysis of Practical Action's Livelihood-Centred*
*Disaster Risk Reduction Project in Nepal Suggested citation*.



World Bank. (2024). *Population growth (annual %) - Kyrgyz Republic*. The World Bank Group. https://data.worldbank.org/indicator/SP.POP.GROW?end=2023&locations=KG&start=2000&view=chart&year=2008%29

Zaginaev, V., Ballesteros-Cánovas, J. A., Erokhin, S., Matov, E., Petrakov, D., & Stoffel, M. (2016). Reconstruction of glacial lake outburst floods in northern Tien Shan: Implications for hazard assessment. *Geomorphology*, *269*, 75–84. https://doi.org/10.1016/j.geomorph.2016.06.028

Zaginaev, V., Falatkova, K., Jansky, B., Sobr, M., & Erokhin, S. (2019). Development of a potentially hazardous pro-glacial lake in Aksay Valley, Kyrgyz Range, Northern Tien Shan. *Hydrology*, *6*(1). https://doi.org/10.3390/hydrology6010003

Zaginaev, V., Petrakov, D., Erokhin, S., Meleshko, A., Stoffel, M., & Ballesteros-Cánovas, J. A. (2019). Geomorphic control on regional glacier lake outburst flood and debris flow activity over northern Tien Shan. *Global and Planetary Change*, *176*(November 2018),
50–59. https://doi.org/10.1016/j.gloplacha.2019.03.003

Zaginaev, V. V., Sakyev, D. J., Nazarkulov, K. B., Amanova, M. T. , Isaev, E. K., & Erokhin, S. A. (2024). Monitoring the Development of Potential Hazardous Mountain Lakes Using Remote Sensing in the Kyrgyz Republic. *International Journal of Geoinformatics*, *20*(12), 36–48. https://doi.org/10.52939/ijg.v20i12.3771

Zhang, G., Carrivick, J. L., Emmer, A., Shugar, D. H., Veh, G., Wang, X., Labedz, C., Mergili, M., Mölg, N., Huss, M., Allen, S., Sugiyama, S., & Lützow, N. (2024). Characteristics and changes of glacial lakes and outburst floods. In *Nature Reviews Earth and Environment* (Vol. 5, Issue 6, pp. 447–462). Springer Nature. https://doi.org/10.1038/s43017-024-00554-w

Zhang, T., Wang, W., & An, B. (2024). A massive lateral moraine collapse triggered the 2023 South Lhonak Lake outburst flood, Sikkim Himalayas. *Landslides*. https://doi.org/10.1007/s10346-024-02358-x