# Peer review of "Modelling the effectiveness of GLOF DRM measures - a case study from the Ala-Archa valley, Kyrgyz Republic"

_EGUsphere, 2025_

## Author Response (AR1)

**Final author comment (AC) as response to referee comments (RC)**

**Referee Comment 1:**

Thank you very much for your detailed and positive feedback. All your comments will be addressed as detailed below, and the manuscript will be revised accordingly.

*Lines 80-85 – I am not pretty sure that 2015 debris flow in Aksay catchment has been caused by the Uchitel Lake burst. Most probably it was a result of the Aksay lake (cavity) burst.*

According to the Ministry of Emergency Situations of the Kyrgyz Republic It was a combination of factors: precipitation and partial outburst from Uchitel Lake. The Kyrgyz Ministry of Emergency visited the glacier after the event and found evidence of outburst channels from Uchitel Lake.

**Suggested text modification:** Lake Uchitel burst in 2015 and – combined with intense rainfall – caused damage downstream, partially destroying a road and several buildings on the Aksay fan, according to field observations of the national Ministry of Emergency Situations (MES). The englacial Aksay lake, that had burst in several prior situations, potentially played a contributing role as well.

*Line 125 (Fig.4) – it will be better to add isolines or cross-sections of the Aksay riverbed add deflection dam to illustrate why this dam should be expanded significantly.*

We will add isolines for illustration and consider adding a cross section of the riverbed, including also the deflection dam.

*Lines 165-170 - The Uchitel Lake is located near the Uchitel Glacier terminus, so projected retreat of the Aksay glacier will not directly cause expansion of the lake. But the Uchitel Glacier will also retreat. A bit more detailed sketch map with both glaciers and lakes (+Aksay) will be useful.*

Thank you. The naming here will be corrected.

*Lines 175-180 - I fully agree with discharge estimates, but values listed in the text are maximum debris flow discharge and are much higher than initial outburst discharge. It is necessary to specify it in the text and explain how these values were used in RAMMS simulation.*

We fully agree. We will add a sentence to make clear that maximum discharge in the RAMMS input parameters refers to the peak discharge values at the lake (highest point of the release hydrograph) while some of the historical values refer to maximum discharge along the flow path. The maximum discharge values that we chose for the initial hydrograph result from both considerations of historical peak discharge values (measured or back-calculated along the flow path) and peak initial discharge calculations based on equations used in the literature (e.g. Huggel et al. 2004; Popov 1991; manuscript references).

*Line 200: - Only flow height for the discrimination of high and medium intensity… It is quite reasonable, but what flow height corresponds to high/medium intensity? Please specify the values.*

Thank you for catching that. We used > 1m flow height for high intensity and $\geq 0.1$ m and < 1 m flow height for medium intensity. These values will be specified in the manuscript.

**Line 295: -** *Fig. 8 it will be better to show location of the deflection dam at the map.*

We will consider adding the location of the deflection dam and the retention basin on this figure.

**Referee Comment 2:**

Thank you very much for your review. We much appreciate your constructive and critical questions. All your comments will be addressed as detailed below, and the manuscript will be revised accordingly.

*Comment 1: I was wondering why GLOFs happened there in the past and why the authors assume they will happen in future again (is there any quazi-cyclic behavior)? Importantly, observed / assumed GLOF triggers and mechanisms should be mentioned in the text as it is essential to understand them in order to choose the most suitable (combination of) measure(s).*

- All outbursts except the one in 2015 were associated with the outburst of an englacial lake on the Aksay glacier. This is described in detail in an article by Zaginaev et al. (2016). The Aksay glacier has significantly degraded over the last 50 years, but the renewed formation of an englacial lake cannot be ruled out completely. Currently, there is a threat of an outburst from neighbouring proglacial Lake Uchitel that was involved in the 2015 GLOF and that has repeatedly been filling in summer, resulting in an overflow for example in 2023.
- Thank you for this comment. We agree and will address the aspect of GLOF triggers and outburst mechanisms in the manuscript.

*Comment 2: The authors assume the lakes can be much larger in future than they are now (large scenario is about 3x max. observed lake volume) and the authors mention quite some variability in observed lake volumes over time; are the lakes growing in the long-term perspective? is there any trend over time? Are there any data supporting assumed future lake extents?*

- We assume that the lakes may grow in the near to mid-term future as the adjacent glaciers will most likely retreat significantly over the next decades which will free space for the lakes to expand (Utchitel/Aksay, Adygene). The current maximum depths measured by bathymetry are at the glacier calving front, indicating that the depression and therefore the lake volume may be increasing overproportionally with glacier retreat. In addition, the depth of the lake may further increase due to thawing and ice melt at the lake bottom.
- For the Aksay valley these assumptions are largely in line with data from Zheng et al. 2021, that indicate an increase of Aksay lake to roughly a volume of 230'000m3 by 2050 under rcp 8.5 and by 2100 under rcp's 2.6-8.5 and in icefree conditions. This fits in between our medium and large scenarios for Uchitel lake (200'000 and 300'000m3 respectively) that is representative for the Aksay valley.
- Zheng et al. 2021 and Furian et al 2021 project no lake at the position of Uchitel lake or Teztor lake, neither at present nor in the future. However, it remains to be said that the used models are possibly not calibrated or suited for the present kind of terrain. From fieldwork we know that there are lakes in both locations and that they have burst and caused damage in past occasions.
- Zheng et al 2021 and Furian et al 2021 additionally indicate the formation of some new lakes for example at Adygene glacier, Golubin glacier, Topkaragay, Toktogul and little western ala-archa lake for which the projected volumes vary quite a lot (Adygene: 30'000m3 and 1.187 M m3; Golubin: 75'000m3 and no lake; Topkaragay: no lake and 250'000m3, Toktogul: no lake and 750'000m3, Little western ala-archa lake: no lake and 110'000m3 + 60'000m3).

- It may be expected that an outburst from one of these additional projected lakes (e.g. at Golubin or Topkaragay glacier) would be in the order of magnitude of our assumed scenarios of the Aksay valley and should cause comparable flow depths.
- Some of the larger values do not match the local observations. For example, a bathymetric survey of the little western ala-archa lake indicates smaller depths than suggested by Furian et al. 2021). Additionally, both the little western ala-archa lake as well as the large Adygene lake are largely bedrock dammed which significantly lowers the probability of a catastrophic outburst. At the projected Toktogul lake (Furian et al 2021: 33m depth) the corresponding glacier has uncovered a flat glacier bed hosting only a small and shallow lake.
- Non-stationary lake formation at Teztor makes it difficult to estimate future volumes. Taking 400,000 $m^3$ (compared to a historical max. estimated at 150,000 $m^3$) is probably the upper limit of potential lake size at this site, but possible, considering the topography, and therfore chosen, also folowing a worst-case approach.
- We will add a paragraph to our manuscript that compares our assumptions and choice of lakes to the data of Zheng et al. 2021 and Furian et al. 2021 and explains the assumed volumes in terms of expected future changes.

**Comment 3**: *Talking about the future – dams and similar man-made structures are usually designed for specified timeframes (longevity), e.g., 100 years; I don't know Kyrgyz guidelines but these are decades or even centuries and this is pretty long period considering glacier dynamics and possibly new lakes in the valley; in order to avoid the need to re-design and re-build the mitigation measures constructed for current lakes; I was wondering if there is any possibility that new lakes will form with continuing ice loss in given timeframes (e.g., in the parallel valley west from Uchitel where the topography suggests possibly large overdeepenings which may (or may not?) be exposed in coming decades)? A discussion of long-term utility value of proposed measures should reflect this aspect (not only possible evolution of existing lakes).*

- The glaciers in the two parallel valleys to the Uchitel glacier can be expected to retreat at similar rates. However, the lower parts of these valleys contain large rock glacier features that give less space for lake accumulation in these possible overdeepenings. Additionally, due to the large area covered with these features, an outburst can be expected to be more gradual (longer distances for the water to cover when draining) and less catastrophic. Aksay glacier (in the west next to Uchitel glacier) could also expose a possibly large overdeepening in the next decades. According to Zheng et al. 2021 and Furian et al. 2021, such an overdeepening could give space for a lake of up to 230'000m3 (Zheng et al. 2021) or 470'000m3 (Furian et al. 2021). Based on these assumptions and on site observations, we think it reasonable to expect that a possible future Aksay lake volume would be of comparable dimensions (in orders of magnitude) to the lake expanding in the parallel Uchitel glacier valley.
- Long-term utility value of the measures: The benefit of all the compared measures is that they would function for a comparable lake in an adjacent valley too. E.g., the deflection dam on the Aksay fan would be effective for a comparable GLOF from Aksay glacier too. The retention basin would interrupt debris flows from any of the other upstream valleys too that could in the future become sources of GLOFs. Similarly, partial and repeated lake drainage offers the advantage of flexibility in case that a lake is being reformed in a newly developed depression in an adjacent valley or adjacent glacier. Further, also 'regular' debris flows, unrelated to lake outbursts, need to be taken into consideration.
- We will add a sentence on this in the discussion part of the manuscript.

***Comment 4***: *It is not clear how peak discharge (a critical input for any GLOF modelling) was estimated in the study? And this goes back to my question about GLOF mechanism (for instance, if the drainage mechanism is opening of sub-surface channels, larger lake volume does not necessarily imply higher peak discharge (it can be limited by the size of this outflow channel, therefore, large lake volume could possibly lead to larger flood volume but comparable peak discharge).*

- Peak discharge was estimated based on the assumption of an outburst through surface breach as is reasonable to assume for a GLOF from Uchitel lake or Teztor lake, based on estimates form historical events and considering empirical relations of lake volume and peak discharge from literture (e.g. Huggel et al. 2004, Popov 1991; cf. response to comment 4 of Reviewer 1). We therefore relate higher volumes to higher peak discharge. We are aware that this may not necessarily hold true for englacial ur sub-surface channels such as in the case of an outburst from englacial Aksay lake. However, in favor of safety and to cover a worst case scenario we assume for the GLOF breach mechanism to be surface rather than subsurface drainage. The changes implemented according the the related comments from Reviewer 1 should clarify this in the manuscript.

***Comment 5***: *The authors argue that the valley experienced larger outbursts in the past, judging from the size of the alluvial fan, but this size of this fan does not suggest larger outbursts in the past; it suggests that a lot of material is being transported from the valley.*

- Agreed. We will adjust the wording and delete the part that associates the size of the fan with the magnitude of the GLOF event. Rather, in the study site section, we wanted to highlight that repeated outbursts in the past have contributed to the formation of the fan (as stated in the literature) and that palaeo-reconstructions showed high debris flow activity in this valley in the past (Zaginaev et al., 2016).

***Comment 6***: *When it comes to the considered hazard mitigation measures, I wonder why complete artificial drainage of the studied lakes is not considered; isn't it easier and cheaper to dig open cut that will prevent lake from filling above certain level instead of repeated draining or bulldozing out hundreds of thousands m^3 of material and enhancing existing dam in sediment-filled retention basin downstream (which will anyway be filled with sediments again)? In other words, the question of interest for DRR is not "what happen if we reduce lake volume by 50%?" but "how much water reduction do we need to achieve "safe" flood (or acceptable risk) levels in case of an outburst?*

- We consider repeated partial drainage rather than complete drainage through an open cut because some of these non-stationary lakes vary in location from year to year. Annual drainage is more flexible in the face of a changing hazard environment associated with glacier change and complex deformation of permafrost terrain.
- This should become clear after the manuscript adjustments that will address some of the former comments.

References

Furian, W., Loibl, D., and Schneider, C. (2021). Future Glacial Lakes in High Mountain Asia: an Inventory and Assessment of hazard Potential from Surrounding Slopes. *J. Glaciol.* 67, 1–18. doi:10.1017/jog.2021.18

Zaginaev, V., Ballesteros-Cánovas, J. A., Erokhin, S., Matov, E., Petrakov, D., & Stoffel, M. (2016). Reconstruction of glacial lake outburst floods in northern Tien Shan: Implications for hazard assessment. *Geomorphology*, *269*, 75-84.

Zheng, G., Allen, S., Bao, A., Ballesteros-Cánovas, J. A., Huss, M., Zhang, G., Li, J., Yuan, Y., Jiang, L., Yu, T., Chen, W., & Stoffel, M. (2021). Supplementary data to: Increasing risk of glacial lake outburst floods from future Third Pole deglaciation (v1.0) [Data set]. Zenodo. https://doi.org/10.5281/zenodo.4477945